# Inhibition of polar actin assembly by astral microtubules is required for cytokinesis

Anan Chen[1], Luisa Ulloa Severino[2], Thomas C. Panagiotou[3], Trevor F. Moraes[1], Darren A. Yuen [2],
Brigitte D. Lavoie[3] & Andrew Wilde [1,3✉]

During cytokinesis, the actin cytoskeleton is partitioned into two spatially distinct actin iso-form specific networks: a β-actin network that generates the equatorial contractile ring, and a γ-actin network that localizes to the cell cortex. Here we demonstrate that the opposing regulation of the β- and γ-actin networks is required for successful cytokinesis. While activation of the formin DIAPH3 at the cytokinetic furrow underlies β-actin filament production, we show that the γ-actin network is specifically depleted at the cell poles through the localized deactivation of the formin DIAPH1. During anaphase, CLIP170 is delivered by astral microtubules and displaces IQGAP1 from DIAPH1, leading to formin autoinhibition, a decrease in cortical stiffness and localized membrane blebbing. The contemporaneous production of a β-actin contractile ring at the cell equator and loss of γ-actin from the poles is required to generate a stable cytokinetic furrow and for the completion of cell division.

[1] Department of Biochemistry, University of Toronto, Toronto, ON, Canada. [2] Keenan Research Centre for Biomedical Science, Li Ka Shing Knowledge Institute, Toronto, ON, Canada. [3] Department of Molecular Genetics, University of Toronto, Toronto, ON, Canada. ✉email: andrew.wilde@utoronto.ca

Cytokinesis is the final stage of the cell cycle when a cell partitions its genetic and cytosolic components into two new cells that then physically separate[1,2]. As each complete copy of the genetic material is segregated to the opposite poles of the cell, a furrow of plasma membrane, the cytokinetic furrow, ingresses between them ultimately fusing and generating two new cells. It follows that an acute spatio-temporal control over mitotic events must be enforced to ensure that the furrow forms at the correct time, after chromosome segregation begins and at the right place, between the segregating chromosomes.

During mitosis the plasma membrane undergoes dramatic remodelling. As a cell enters mitosis it rounds up, a process driven by the remodelling of the actin cytoskeleton to generate a plasma membrane with uniformly higher stiffness[3,4]. As the cell exits mitosis, it elongates and an actomyosin ring, the contractile ring, forms at the cell equator between the segregating chromosomes[1,2]. Consequently, plasma membrane stiffness becomes asymmetric as there is greater tension at the site of furrow ingression[3]. In metazoans, the mechanisms underlying the breaking of symmetry within the plasma membrane that lead to cytokinetic furrow ingression have been hotly debated for decades. The different models focus on the primary site of activity on the plasma membrane required for furrow ingression. In the pole-centric view, furrow ingression is proposed to result from membrane relaxation at the cortical poles of the cell while membrane tension is maintained at the equator[5]. The observation that microtubules position contractile forces at the equatorial site of furrow ingression challenged this view[6], and led to an equator-centric model that has gained widespread support. In this model, microtubules of the central spindle position a region of active RhoA-GTP at the adjacent plasma membrane that directs the assembly of a contractile actomyosin ring where furrow ingression occurs[7]. The contractile ring is built by the de novo assembly of β-actin filaments at the cell equator through the action of the formin, DIAPH3[8–10]. DIAPH3 is recruited to the cytokinetic furrow as the cell exits metaphase by the scaffolding protein anillin, and requires activation through the binding of both RhoA and anillin to release it from its autoinhibited state[9]. Strikingly, we found that DIAPH3 preferentially nucleates β-actin filaments, suggesting that the cytokinetic ring is comprised of a specialized β-actin filament network that is primarily regulated by DIAPH3 activity. While a greater focus has been on events at the cell equator during anaphase, other studies have pointed to the importance of cytokinetic events at the cell poles[11–14]. During anaphase, the actin cytoskeleton partitions into discrete actin-isoform networks, with β-actin enriched at the cell equator and γ-actin around the cortex[8–10]. Importantly, the localized disruption of actin at the pole or the equator has different effects on cytokinesis[12], suggesting that the different actin networks perform specialized functions at the poles and the equator. However, while our understanding of the molecular pathways that generate and organize the β-actin equatorial cytokinetic network has progressed[9], little is known of the nucleators and organizers of the cytokinetic cortical γ-actin network.

During anaphase, furrow ingression at the equator increases membrane tension resulting in greater internal hydrostatic pressure within the cell[3,13]. If the hydrostatic pressure is unequal between the two halves of the cell, the furrow is destabilized and cytokinesis fails[13]. To alleviate any pressure imbalance during anaphase, small membranous extrusions, termed blebs, form at the cell poles[13]. These and other observations demonstrate that events at the poles of the cell are important for successful cytokinesis[11,15,16].

Here, we investigate the roles of distinctly localized β- and γ-actin isoform networks in furrow ingression during mammalian cytokinesis. We identify a direct pathway whereby astral microtubules deliver a signal to the cell poles during anaphase, abrogating local γ-actin filament nucleation and constraining blebbing to the poles. Finally, we demonstrate that the anaphase remodeling of the actin cytoskeleton into distinct β- and γ-actin networks ensures efficient furrow ingression, and derives from the coordinated counter regulation of actin nucleators, notably the activation of DIAPH3 at the cell equator and the deactivation of DIAPH1 at the cell poles, which are both required for the successful completion of cytokinesis.

## Results

### The cortical γ-actin network depends on DIAPH1 and is required for cytokinesis.
During anaphase, the actin cytoskeleton partitions into distinct actin-isoform networks (Fig. 1a, Supplementary Fig. 1a)[8,9]. During metaphase, β and γ-actin are equally distributed around the cell cortex. However, as the chromosomes segregate during anaphase A, and prior to furrow ingression, β and γ-actin begin to asymmetrically redistribute around the cortex such that β-actin becomes enriched at the site of future furrow ingression at the cell equator. In contrast, γ-actin remains distributed around the cortex, becoming the dominant actin network at the cell poles. As anaphase progresses and furrowing begins, the asymmetry of actin-isoform distribution increases (Supplementary Fig. 1a). This asymmetry of the two different actin networks is all the more remarkable given that β-actin differs from γ-actin by only four amino acids[8,9]. To understand how the polar γ-actin network functions in cytokinesis, we first sought to determine how it is generated.

In metaphase DIAPH1, DIAPH3, β-actin, and γ-actin are uniformly distributed around the cortex. However, during anaphase A prior to furrowing, the formin DIAPH3 becomes enriched at the cell equator where it specifically nucleates the β-actin filaments that stabilize furrow ingression (Fig. 1a)[9]. In contrast, the closely related formin DIAPH1 localizes to the cell cortex during anaphase along with γ-actin filaments (Fig. 1a), suggesting that DIAPH1 could nucleate the cortical actin network. We assessed the role of DIAPH1 and γ-actin filaments in cytokinesis by siRNA depletion, which caused an increase in the number of multinucleated cells, a hallmark of cytokinetic failure (Fig. 1b). Expression of a siRNA resistant GFP-DIAPH1 rescued the multinucleate phenotype, whereas expression of the DIAPH1-I862A mutant, which cannot nucleate actin filaments[17], did not (Fig. 1b, Supplementary Fig. 1b). We next assessed the integrity of the β- and γ-actin-isoform networks in cells depleted for DIAPH1 or IQGAP1, a DIAPH1 interactor that recruits DIAPH1 to the cell cortex and functions as an enhancer of its actin nucleation activity[18]. siRNA depletion of either DIAPH1 or IQGAP1 reduced the amount of γ-actin at the cortex, but did not disrupt β-actin assembly at the cell equator (Fig. 1c, Supplementary Fig. 1c, d). As depletion of DIAPH1 did not cause a redistribution of DIAPH3 to the poles nor did DIAPH3 depletion cause a redistribution of DIAPH1 to the equator (Supplementary Fig. 1e), these data demonstrate that the β- and γ-actin cytoskeletons are independently generated by specific formins.

### The DIAPH1 cortical γ-actin network restricts blebs to the cell poles.
To determine the cellular role of DIAPH1 and the cortical γ-actin network during cytokinesis, we performed phase contrast time-lapse imaging of DIAPH1 siRNA treated cells. In control dividing cells there is an increase in the appearance of small dynamic protrusions of the plasma membrane (termed blebs) that are predominantly restricted to the poles of anaphase cells (Fig. 1d, Supplementary movie 1). In contrast, cells depleted of DIAPH1 exhibited a significant increase in the number of blebs that now occurred all over the plasma membrane (Fig. 1d, e,

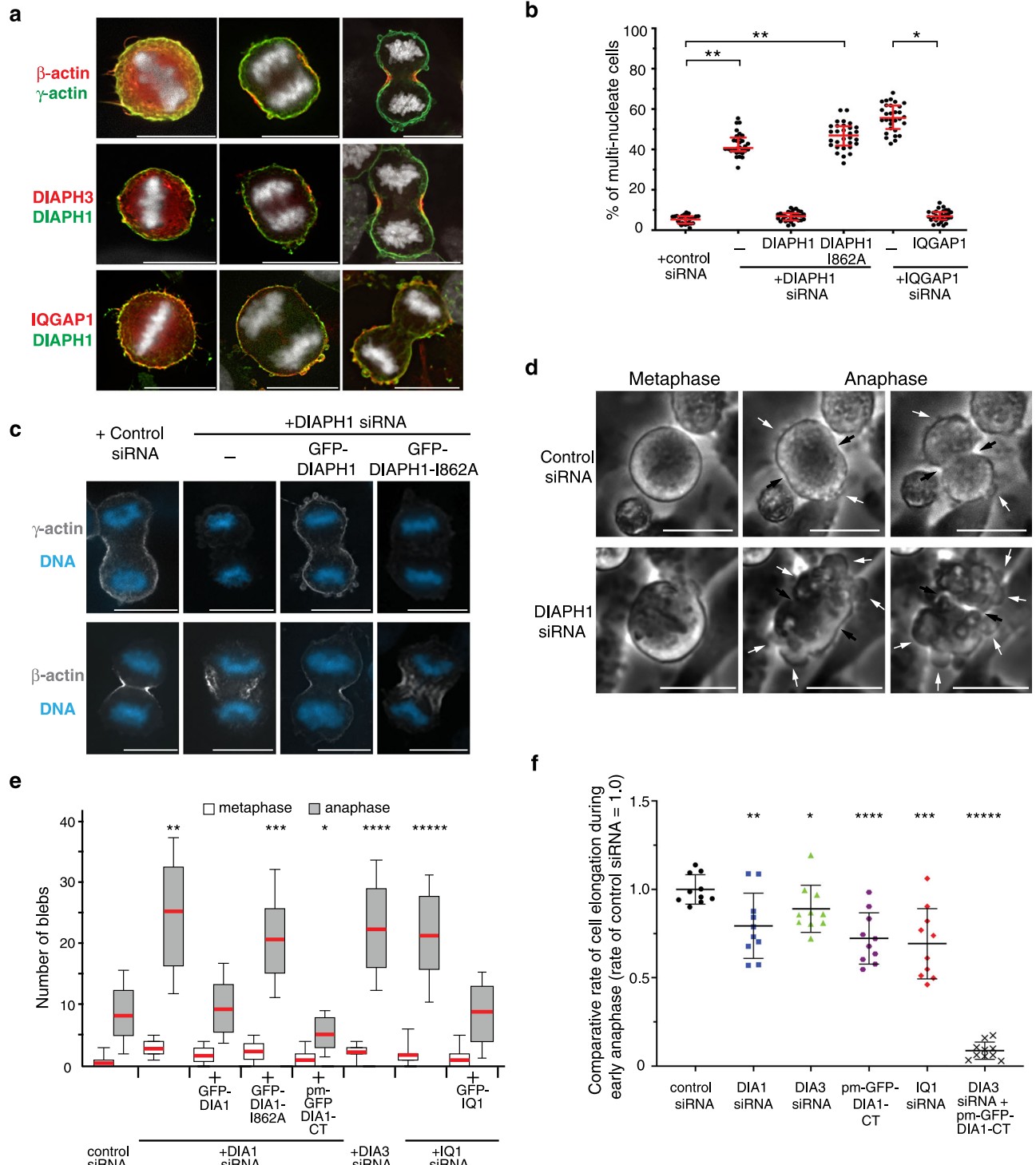

Supplementary movie 2), suggesting that DIAPH1 functions to constrain blebs to the poles of cells during cytokinesis. In addition, the rate of cell elongation during anaphase B was reduced in DIAPH1-depleted cells (Fig. 1f, Supplementary Fig. 2a–c). We quantified cell elongation in both live and fixed cells. Live imaging analysis revealed that disruption of the γ or β-actin networks generated by DIAPH1 or 3, respectively, reduced the rate of cell elongation, but that cells eventually elongated to a length comparable with control cells (Fig. 1f, Supplementary Fig. 2a). Analysis of anaphase DIAPH1-depleted fixed cells with similarly separated chromosomes masses (8 ± 0.29 μm apart) or with comparable furrow depths concurred with the live-cell imaging

(Supplementary Fig. 2b, c). Importantly, cell elongation defects were rescued upon expression of GFP-tagged full-length DIAPH1, but not a DIAPH1 mutant unable to nucleate actin filaments, DIAPH1-I862A (Supplementary Fig. 2b), indicating that the regulation of the γ-actin cortical network is involved in cell elongation.

Both the delocalized and increased plasma membrane blebbing observed in DIAPH1-depleted cells are suggestive of a loss of global cortical stiffness. To test this directly, we used Atomic Force Microscopy to measure cortical stiffness (Fig. 2 and Supplementary Fig. 3). Due to the dramatic increase in cell blebbing observed during anaphase in DIAPH1-depleted cells,

**Fig. 1 Cortical γ-actin is required for cytokinesis. a** Dividing HeLa cells fixed and stained with antibodies recognizing β-actin, γ-actin, DIAPH1, DIAPH3, and IQGAP1. **b** Quantitation of the number of multinucleated cells in the presence and absence of DIAPH1, IQGAP1, and DIAPH1-I862A that cannot nucleate actin. Bars denote the average, whiskers denote ±SD. *$p = 0.0001$, **$p < 0.0001$ to control using nonparametric two-tailed Mann-Whitney $t$-tests for three experimental repeats, $n = 600$ cells examined over three independent experiments. **c** β-actin or γ-actin localization in HeLa cells treated with control siRNA or DIAPH1 siRNA in the presence and absence of exogenously expressed GFP-DIAPH1 or GFP-DIAPH1-I862A. **d** Frames from time-lapse phase contrast movies demonstrating that depletion of DIAPH1 caused an increase in plasma membrane blebbing during anaphase. White arrows denote, blebs, black arrows point to the cytokinetic furrow. **e** Quantitation of the number of plasma membrane blebs in fixed metaphase and anaphase cells. Bars denote the average, boxes represent 25–75 percentile and whiskers denote extremes. *$p = 0.014$, **$p = 0.0007$, ***$p = 0.0006$, ****$p = 0.0005$, *****$p = 0.0004$ difference to control using nonparametric two-tailed Mann-Whitney $t$-tests for three experimental repeats, $n = 300$ cells examined over three independent experiments. CT = C terminus; DIA1 = DIAPH1; DIA3 = DIAPH3; IQ1 = IQGAP1. **f** Quantitation of the rate of cell elongation from 2–8 min after anaphase onset. Bars denote the average, whiskers denote ±SD. *$p = 0.029$, **$p = 0.005$, ***$p = 0.0007$, ****$p = 0.0003$, *****$p < 0.0001$ to control, using nonparametric two-tailed Mann-Whitney $t$-tests for three experimental repeats, $n = 10$ cells examined in each live-cell imaging group. CT = C terminus; DIA1 = DIAPH1; DIA3 = DIAPH3. Scale bars are 10 μm.

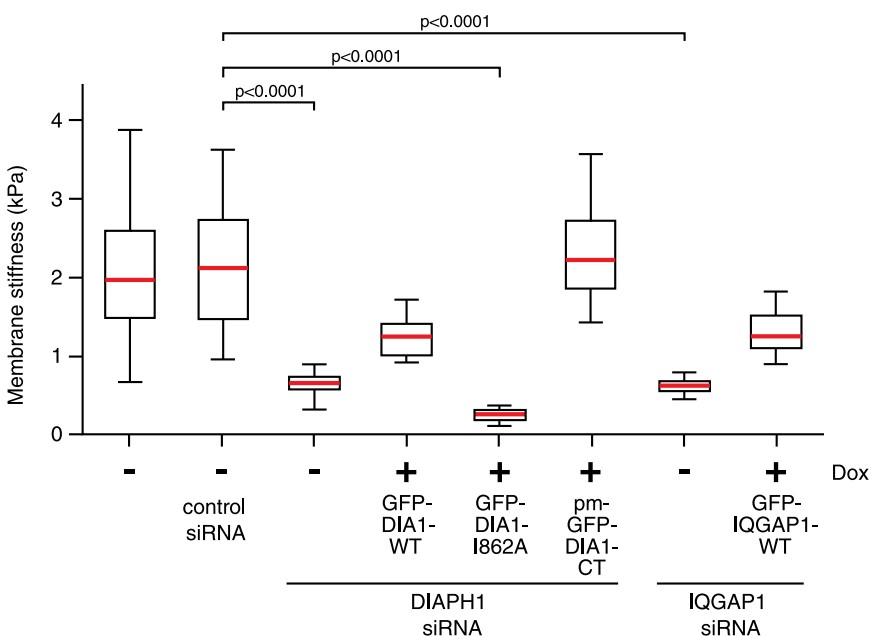

**Fig. 2 Cortical γ-actin is required for metaphase cortical stiffness.** Young's modulus, measured in kPa, of HeLa cells in metaphase was measured by Atomic Force Microscopy. Actual numerical values and force curves are shown in Supplementary Fig. 3d. $p < 0.0001$, using nonparametric two-tailed Mann-Whitney $t$-tests for three experimental repeats. $n = 29, 29, 29, 27, 29, 29, 27$, or 28 cells examined over three independent experiments (correlated to each condition from the left to the right in the graph). Bars = average, boxes = 25–75 percentile, whiskers = extreme. DIA1 = DIAPH1, CT = C-terminus, WT = wildtype, DOX = Doxycycline.

measurements were restricted to metaphase cells. In metaphase, DIAPH1 depletion caused a 3-fold reduction in the modulus of elasticity (Young's modulus), demonstrating a loss of cortical stiffness that was restored in cells expressing the constitutively active DIAPH1-CT fragment (pm-GFP-DIA1-CT). Similarly, expression of full-length GFP-DIAPH1, which requires activation to release it from its basal autoinhibited state, partially rescued membrane stiffness, while no rescue was observed in cells expressing the DIAPH1-I862A mutant. These data suggest that DIAPH1 actin nucleation activity contributes to maintain cortical stiffness.

**Opposing regulation of the γ- and β-actin cytoskeletal networks is required for furrow ingression.** To assess cortical actin dynamics during anaphase and determine how they contribute to successful cytokinesis, we used time-lapse fluorescence microscopy to follow the actin filament reporter Lifeact-GFP in dividing cells. As anaphase proceeded, we observed that the Lifeact-GFP signal decreased at the cell poles (Fig. 3a). To determine if the loss of polar actin is required for cytokinesis, we

next targeted constitutively active DIAPH1-CT to the plasma membrane (pm-GFP-DIA1-CT) to continually produce γ-actin filaments at the cell cortex throughout mitosis. Under these conditions, polar membrane blebbing was reduced, the number of multinucleated cells increased and anaphase cell elongation rates dropped (Fig. 1e, f), indicating that anaphase remodeling of the cortical actin network is required for cytokinesis, and that the presence of constitutively active DIAPH1 at the cell poles interferes with this process.

While either the loss or the constant generation of cortical γ-actin filaments at the cortex prevented the completion of cytokinesis, furrow ingression at the cell equator occurred (Fig. 3b, c, Supplementary Fig. 2, Supplementary movies 2 and 3, respectively) as the β-actin rich contractile ring still assembles at the cell equator[9]. Furrow ingression dynamics were, however, affected: the sustained production of cortical γ-actin filaments in cells expressing pm-GFP-DIA1-CT slowed the rate of furrow ingression to $66 \pm 0.12\%$ of that of control cells (Fig. 3b, Supplementary Fig. 2d). Interestingly, the loss of the β-actin contractile ring upon depletion of either DIAPH3 or anillin, an enhancer of DIAPH3, also allowed a rudimentary, although

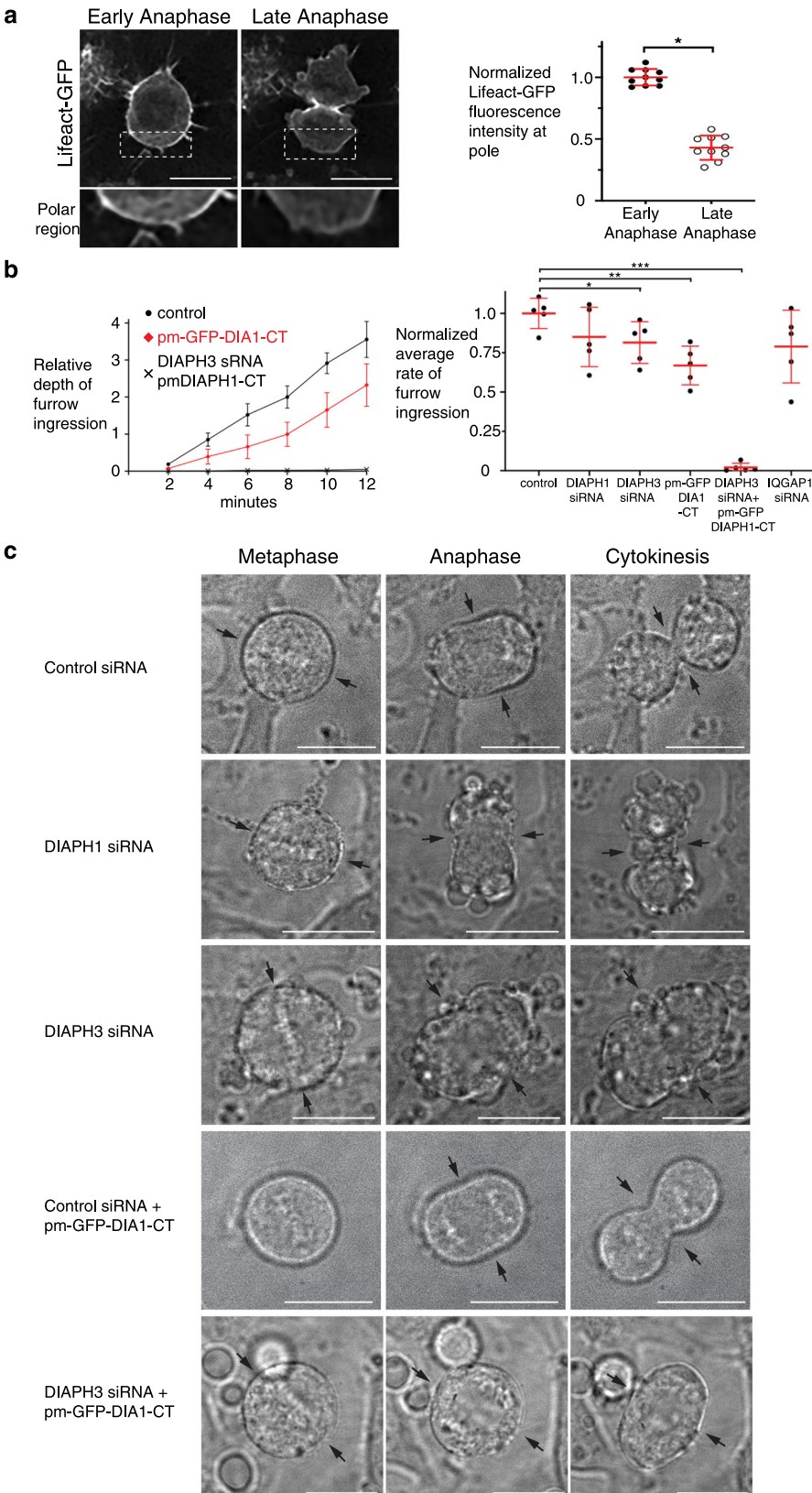

unstable furrow to ingress (Fig. 3b, c, Supplementary Fig. 2, Supplementary movie 4)[9,13,19]. Ingression of the rudimentary furrow was not due to the repositioning of the DIAPH1 formin to the furrow in the absence of DIAPH3 (Supplementary Fig. 1e). To determine if the furrow formed in the absence of β-actin is dependent on the γ-actin depletion at the poles, we assessed

furrow ingression in cells where normal actin-isoform dynamics and distribution were disrupted (Fig. 3b, c). DIAPH3 was depleted, thereby abrogating equatorial β-actin filament production, whilst a constitutively active DIAPH1 (pm-DIA1-CT) was targeted to the plasma membrane to sustain the continuous production of cortical γ-actin filaments throughout anaphase.

**Fig. 3 Opposing actin-isoform dynamics between the cell pole and equator are required for cytokinetic furrow ingression and successful cell division. a** Lifeact-GFP signal diminishes at the poles of dividing cells. Boxed region is magnified in the lower panels and the Lifeact-GFP signal on the cortex within the boxed region was quantified. Bars denote the average, whiskers denote ±SD, *p = 0.0003 using nonparametric two-tailed Mann-Whitney t-tests. n = 10 cells examined in each live-cell imaging group. **b** Relative depth of furrow ingression over time and the comparative rate of furrow ingression in different conditions. *p = 0.03, **p = 0.008, and ***p < 0.0001 using nonparametric two-tailed Mann-Whitney t-tests. n = 5 cells examined in each live-cell imaging group. Dots = average (left panel), bars = average (right panel); whiskers denote ±SD. DIA1 = DIAPH1, CT = C-terminus. **c** Frames from phase contrast time-lapse series in cells depleted of DIAPH1 or DIAPH3 in the presence or absence of expressed constitutively active DIAPH1 targeted to the plasma membrane (pm). Arrows point to cell equator. Scale bar is 10 μm.

These conditions abrogate polar relaxation, as judged by the absence of bleb formation during anaphase (Fig. 3c, Supplementary movie 5), and only one out of eight cells formed a furrow. In contrast, most cells depleted of either DIAPH1 (Supplementary movie 2) or DIAPH3 (Supplementary video 4), or expressing constitutively active DIAPH1 in the presence of normal β-actin dynamics (Supplementary movie 3), were able to generate a furrow (9 of 10, 8 of 10 and 9 of 10 cells, respectively, Fig. 3b, c). Thus, in the absence of the β-actin contractile ring, polar relaxation is sufficient to promote rudimentary furrow formation, but fails to support cytokinesis. These data demonstrate that the contemporaneous assembly of an equatorial β-actin contractile ring along with the depletion of the polar γ-actin network coordinately orchestrate cytokinesis and moreover suggest that distinct mechanisms must independently regulate each actin-isoform network.

**Astral microtubules are required for polar γ-actin depletion.** During anaphase, stabilized astral microtubules enhance polar blebbing[15,20], suggesting that astral microtubules could deliver factors that negatively regulate the γ-actin network at the poles. To specifically disrupt astral, but not spindle, microtubules, we treated cells with the PLK4 inhibitor Centrinone B that prevents centrosome duplication[21]. Upon replating, Centrinone B-treated cells enter mitosis and build a bipolar spindle where only one spindle pole contains a centrosome (Aurora-A positive) from which astral microtubules emanate, while the other spindle pole lacks a centrosome (Aurora-A negative) and consequently cannot nucleate astral microtubules (Fig. 4a). We observed an increase in cortical γ-actin (Fig. 4b, c) coupled with a decrease in the number of blebs (Fig. 4d) at the cell pole lacking astral microtubules (centrosome minus) relative to that observed at the opposite cell pole with a centrosome and astral microtubules, suggesting that astral microtubules deliver a signal capable of regulating actin dynamics.

**Deactivation of DIAPH1 requires the loss of RhoA and IQGAP1 binding.** For efficient polar actin depletion to occur, the cortical flow of γ-actin from the poles to the equator must also be accompanied by the simultaneous inhibition of new actin filament production. We hypothesized that DIAPH1 could be specifically switched off at the cell poles during anaphase. Previously, we identified a mechanism to activate DIAPH1: the sequential binding of RhoA then IQGAP1 to the N-terminal regulatory domains of DIAPH1 (G-protein binding domain, GBD, and DAD-interacting domain, DID, respectively) releases the actin nucleating DIAPH1 C-terminus from its autoinhibited state, enabling it to generate actin filaments[22]. How activated formins return to their autoinhibited state is unknown. To gain insight into if and how the DIAPH1 actin nucleating activity is switched off during anaphase, we assessed the anaphase localization of DIAPH1, RhoA, and IQGAP1. As expected, RhoA became concentrated at the furrow and was largely absent from the cell poles (Fig. 5a), while DIAPH1 and IQGAP1 colocalized throughout the

cell cortex during anaphase (Fig. 1a). While the initial IQGAP1 binding to DIAPH1 requires RhoA[18] it is not known if IQGAP1 binding to DIAPH1 persists in the absence of RhoA, and if it does, is DIAPH1 maintained in the open active state? To address this question, we performed a series of in vitro biochemical binding assays. To determine if IQGAP1 binding to DIAPH1 in the absence of RhoA maintains the formin in the active state, we assessed the interaction between the DIAPH1 N-terminal regulatory and the C-terminal actin nucleating domains by pull-down assays (Fig. 5b). A recombinant C-terminal half of DIAPH1 (MBP-DIAPH1-CT) can bind to an N-terminal fragment of DIAPH1 (His$_6$-DIAPH1-NT), recapitulating the DIAPH1 auto-inhibited state; however, formin autoinhibition was precluded by preincubation of the His$_6$-DIAPH1-NT with either GST-RhoA, MBP-IQGAP1 or both (Fig. 5b). Importantly, these data indicate that IQGAP1 binding is sufficient to maintain formin activity and that both RhoA and IQGAP1 must dissociate for DIAPH1 to return to its autoinhibited, OFF state.

**CLIP170 removes IQGAP1 from DIAPH1.** As astral microtubules have been implicated in polar relaxation during anaphase, one straightforward possibility is that γ-actin depletion from the cell poles could involve the delivery of a DIAPH1 inhibitor to the cell cortex by microtubules. A candidate effector of γ-actin dynamics is the microtubule associated cytoplasmic linker protein 170 (CLIP170) as it moves through the cell on the plus ends of microtubules[23] and binds to the DBR domain of IQGAP1[24], the same domain that interacts with DIAPH1[25]. To test whether CLIP170 can displace IQGAP1 from DIAPH1, we first measured the dissociation constants of pairwise interactions using biolayer interferometry (BLI). We detected no interaction between DIAPH1 and CLIP170, but found that IQGAP1 has a 4-fold greater affinity for CLIP170 than for DIAPH1 (K$_d$ = 20.5 ± 4.4 nM, K$_d$ = 81.2 ± 9.2 nM[22], respectively, Supplementary Fig. 4), suggesting that CLIP170 could displace IQGAP1 from DIAPH1. To address this directly, we next performed in vitro competition binding assays. Preformed complexes of His$_6$-DIAPH1-NT plus GST-IQGAP1-DBR were incubated with an equimolar amount of MBP-CLIP170-NT. Consistent with the BLI data, CLIP170 efficiently displaced IQGAP1 from the DIAPH1-autoinhibitory domain, thereby promoting DIAPH1 inactivation. In contrast, preformed GST-IQGAP1-DBR plus MBP-CLIP170-NT complexes were not disrupted by the addition of His$_6$-DIAPH1-NT (Fig. 6).

**CLIP170 associates with polar IQGAP1 during anaphase.** For CLIP170 to function as an anaphase OFF switch for DIAPH1, it must be delivered to the polar cortex during anaphase where it can interact with IQGAP1. To determine when and where CLIP170 localizes, we monitored GFP-CLIP170 by time-lapse fluorescence microscopy. As expected, GFP-CLIP170 was concentrated on spindle microtubules in metaphase[26,27], however little or no GFP-CLIP170 was observed on metaphase astral microtubules (Fig. 7a). In contrast, from the onset of anaphase

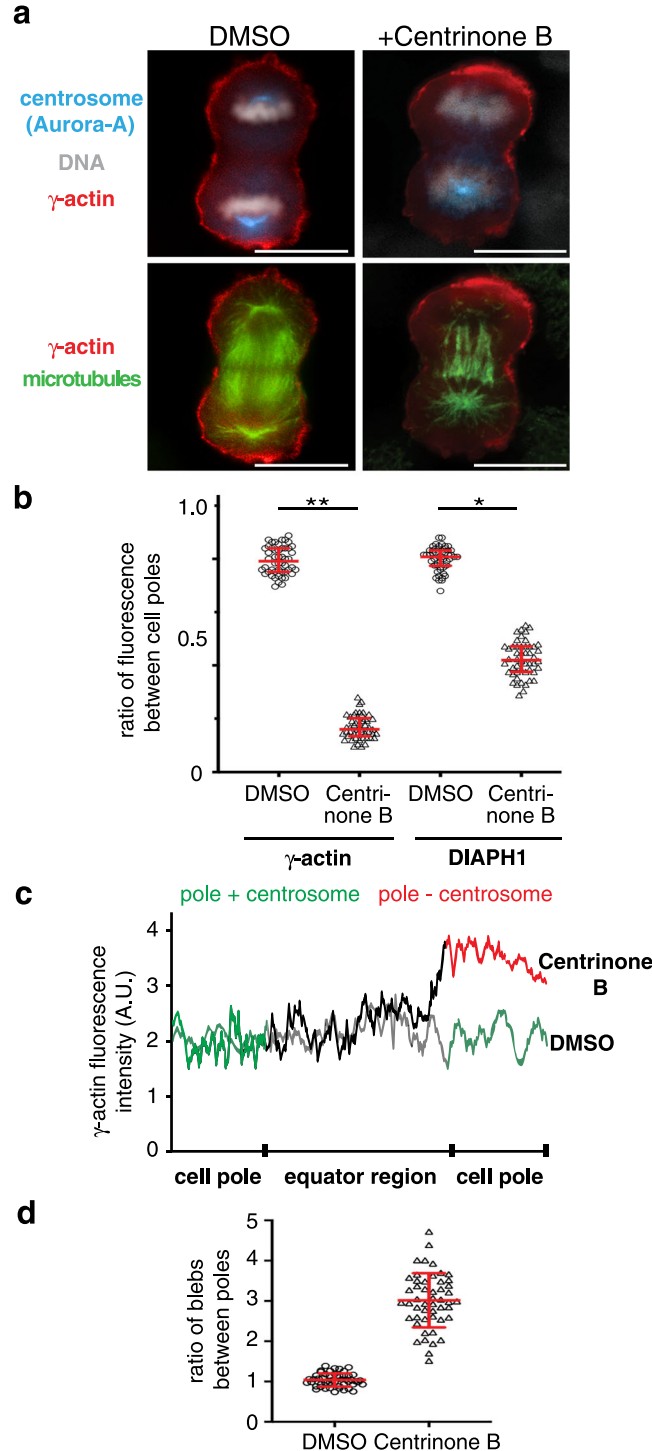

**a**

DMSO    +Centrinone B

centrosome (Aurora-A)
DNA
γ-actin

γ-actin
microtubules

**b**

ratio of fluorescence between cell poles

DMSO  Centrinone B   DMSO  Centrinone B

γ-actin    DIAPH1

**c**

pole + centrosome    pole - centrosome

γ-actin fluorescence intensity (A.U.)

Centrinone B

DMSO

cell pole    equator region    cell pole

**d**

ratio of blebs between poles

DMSO Centrinone B

**Fig. 4 Astral microtubules are required for polar γ-actin clearance during anaphase. a** HeLa cells fixed and probed with antibodies recognizing tubulin (green), γ-actin (red), Aurora-A (blue), and DAPI to stained chromosomes in white. Cells treated with Centrinone B lack a centrosome at one spindle pole (absence of Aurora-A) and astral microtubules. At this pole cortical γ-actin is maintained during anaphase. **b** Ratio of γ-actin and DIAPH1 fluorescence intensity between poles of cells (+astral microtubules:−astral microtubules). Bars denote the average, whiskers denote ±SD. $**p < 0.0001$, $*p = 0.0004$ using nonparametric two-tailed Mann-Whitney $t$-tests for three experimental repeats, $n = 50$ cells examined over three independent experiments. **c** Pole to pole line-scans along the plasma membrane of representative γ-actin stained cells. One cell is DMSO treated and has a centrosome at each spindle pole, the other is Centrinone B-treated with a centrosome at only one spindle pole. **d** Cells poles lacking astral microtubules have fewer plasma membrane blebs. Bars denote the average, whiskers are ±SD for three experimental repeats, $n = 50$ cells examined over three independent experiments. Scale bars are 10 μm.

CLIP170 depleted cells remained in metaphase and failed to progress into anaphase, reflecting CLIP170's vital role in modulating microtubule attachment[26,27]. However, our model predicts that as anaphase advances and DIAPH1 γ-actin filament nucleation activity is progressively deactivated, there should be a reduction in DIAPH1-IQGAP1 interactions and a corresponding increase in IQGAP1-CLIP170 binding. To test this prediction, we used a proximity ligation assay (PLA) that generates fluorescent foci when two epitopes are optimally 10 nm apart, thus inferring an interaction. PLA foci were quantified in cells probed for DIAPH1 and IQGAP1, DIAPH1 and CLIP170, and IQGAP1 and CLIP170. In metaphase, DIAPH1-IQGAP1 PLA foci were abundant around the whole cortex, but as anaphase progressed, the number of foci decreased especially at the poles of cells (Fig. 7b, c). In contrast, IQGAP1-CLIP170 PLA foci were few in metaphase, but increased dramatically as anaphase progressed, again most markedly at the cell poles (Fig. 7b, c). These data position CLIP170 at the correct location and time to deactivate DIAPH1 via disrupting the DIAPH1-IQGAP1 interaction. We have previously demonstrated that IQGAP1 is specifically involved in the activation of DIAPH1 and not DIAPH3[18], which utilizes anillin as its enhancer rather than IQGAP1[9]. While we could detect co-precipitation of IQGAP1 and CLIP170, we detected no co-precipitation of anillin and CLIP170 nor DIAPH3 and CLIP170 (Supplementary Fig. 5). Furthermore, we observed no colocalization of DIAPH3 and CLIP170 in proximity ligation assays (Supplementary Fig. 5). Combined these data suggest that CLIP170 is involved in deactivating DIAPH1 and not DIAPH3.

To determine if the anaphase astral microtubules delivered CLIP170 to the poles, we next assessed PLA foci in Centrinone B-treated cells where astral microtubules are restricted to just one spindle pole. At the astral microtubule positive pole, DIAPH1-IQGAP1 PLA foci decreased during anaphase while IQGAP1-CLIP170 PLA foci increased (Fig. 7d, e). In contrast, at the opposite cell pole lacking astral microtubules, DIAPH1-IQGAP1 PLA foci persisted throughout anaphase and few IQGAP1-CLIP170 PLA foci were observed (Fig. 7d, e). These data show that while DIAPH1 and IQGAP1 interact in metaphase, the degree of interaction diminishes as anaphase progresses when CLIP170 is delivered to the poles by astral microtubules. Correspondingly, IQGAP1-CLIP170 interactions exhibit the opposite dynamics: while few interactions are observed in metaphase, IQGAP1-CLIP170 proximity increases dramatically as anaphase progresses, and this proximity is dependent upon astral microtubules. This proximity data places CLIP170 at the correct place, the polar cortex, at the right time, anaphase, to

onwards, GFP-CLIP170 was observed in astral microtubules where it began to transit towards the polar cortex (Fig. 7a, Supplementary movie 6). In addition, GFP-CLIP170 was observed on spindle midzone microtubules formed between the segregating chromosomes. Immunostaining of fixed HeLa cells expressing GFP-CLIP170 with antibodies recognizing DIAPH1, 3 and IQGAP1 revealed that CLIP170 positive microtubules grew toward polar regions of the cell where DIAPH1 and IQGAP1 localized but few CLIP170 positive microtubules were observed in the equatorial cortex region of the cell where DIAPH3 was enriched in prefurrowing cells (Supplementary Fig. 5). It was not possible to deplete CLIP170 and assess its role in anaphase, as

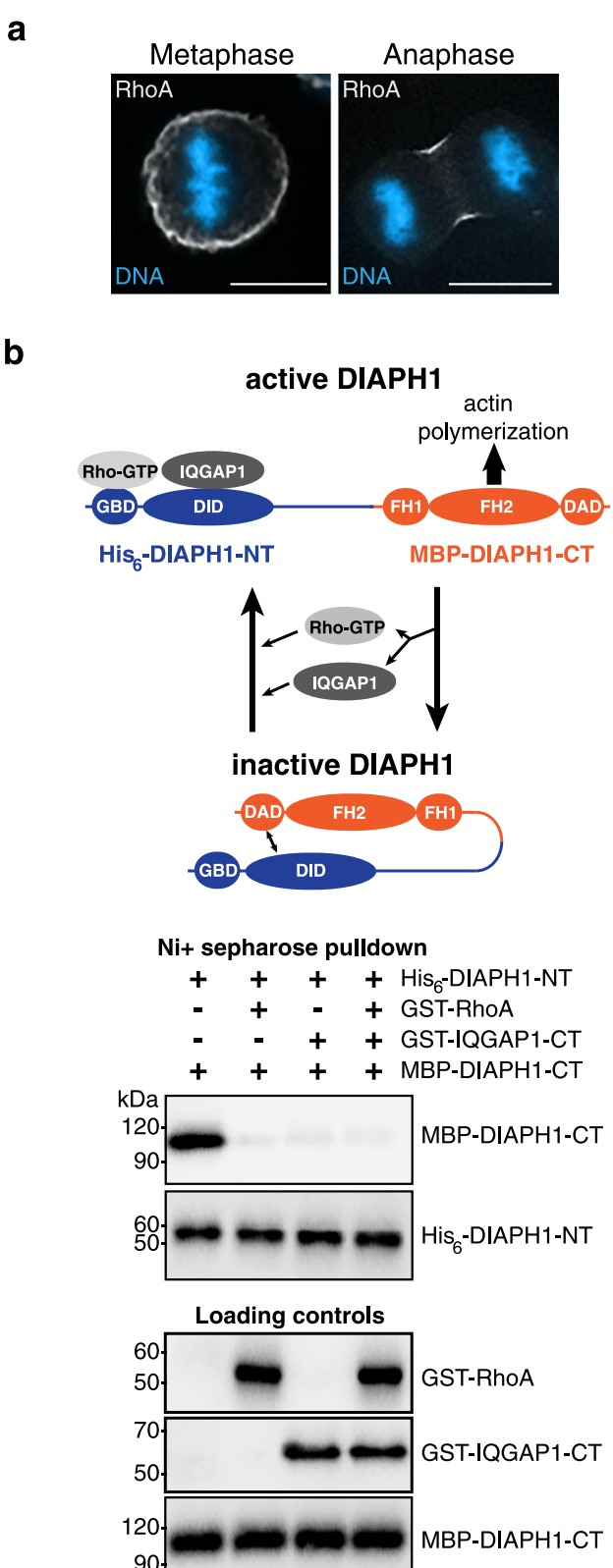

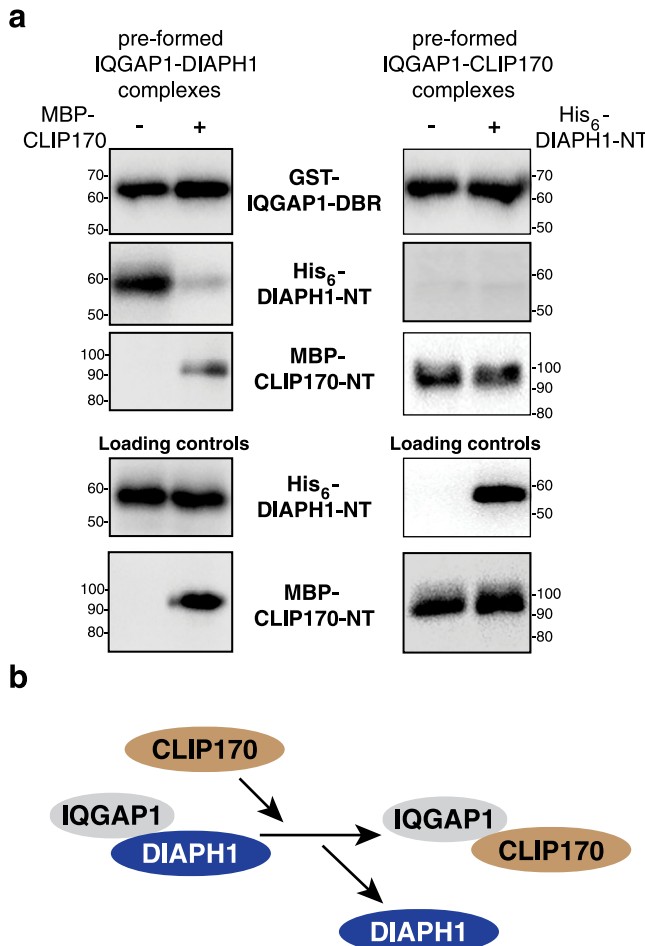

**Fig. 6 IQGAP1 preferentially binds to CLIP170 over DIAPH1. a** Preassembled GST-IQGAP1-DBR and His₆-DIAPH1-NT complexes were incubated with MBP-CLIP170-NT. In parallel preassembled GST-IQGAP1-DBR and MBP-CLIP170-NT complexes were incubated with His₆-DIAPH1-NT. After reisolating GST-IQGAP1-DBR, the amount of co-purifying MBP-CLIP170-NT or His₆-DIAPH1-NT was determined. NT = N-terminus, DBR = Diaphanous Binding Region. **b** Schematic representing inferred IQGAP1 interaction dynamics.

interact with IQGAP1 and remove it from DIAPH1 thereby allowing DIAPH1 to return to the autoinhibited state and be unable to replenish the polar γ-actin removed by cortical flow.

**Discussion**

Our study demonstrates that during mammalian cell anaphase, the two constitutively expressed β- and γ-actin isoforms are partitioned into two physically distinct cytoskeletal networks. Each network is required for efficient furrow ingression and successful cytokinesis. The equatorial β-actin network is generated by the specific targeting and activation of the formin DIAPH3 to the site of future furrow ingression by the combined action of an activator, RhoA and an activity enhancer, anillin[9]. The DIAPH3 dependent generation of β-actin underlies contractile ring assembly and is required to stabilize the cytokinetic furrow[9]. The cortical γ-actin network is generated through similar molecular principles; the formin DIAPH1 is targeted to the cortex by an enhancer of its activity IQGAP1[18]. The cortical γ-actin network regulates cortical membrane stiffness, furrow positioning[12], and ingression kinetics.

**Fig. 5 Deactivation of DIAPH1 requires loss of binding to RhoA and IQGAP1. a** HeLa cell fixed and probed with an antibody recognizing RhoA, white and DAPI stained chromosomes in blue. **b** Preformed complexes of His₆-DIAPH1-NT with GST-RhoA, GST-IQGAP1 or both prevent MBP-DIAPH1-CT binding to His₆-DIAPH1-NT. Schematic depicts that DIAPH1 deactivation requires the removal of both RhoA and IQGAP1. CT = C-terminus.

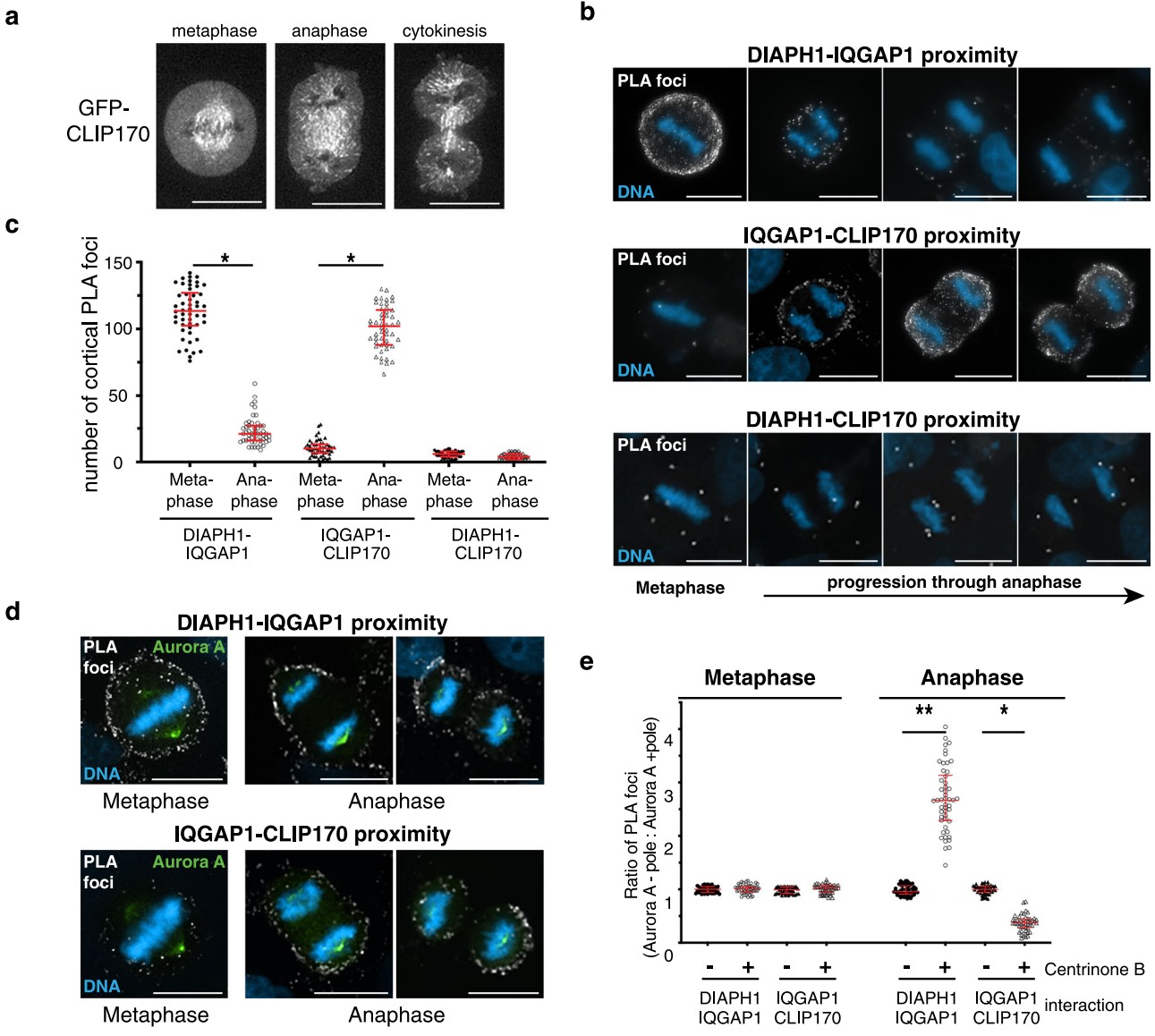

**Fig. 7 The CLIP170-IQGAP1 interaction is dependent upon anaphase astral microtubules. a** Time-lapse images of GFP-CLIP170 during cell division. **b** Proximity ligation assays (PLA) mapping the proximity DIAPH1 to IQGAP1 (top), IQGAP1 to CLIP170 (middle) and CLIP170 to DIAPH1 (bottom) during mitotic progression. **c** Quantitation of PLA foci in metaphase and anaphase *$p < 0.0001$ using a nonparametric two-tailed Mann-Whitney $t$-tests for three experimental repeats, $n = 50$ cells examined over three independent experiments. **d** Proximity ligation assays (PLA) mapping the proximity DIAPH1 to IQGAP1 and IQGAP1 to CLIP170 in Centrinone B-treated cells. **e** Quantitation of PLA foci in **d**. At cell poles lacking astral microtubules, DIAPH1 and IQGAP1 are in close proximity but IQGAP1 and CLIP170 are not. *$p = 0.0006$, **$p < 0.0001$ using nonparametric two-tailed Mann-Whitney $t$-tests for three experimental repeats, $n = 50$ cells examined over three independent experiments. Scale bars are 10 µm.

Both actin networks, DIAPH1, DIAPH3, and RhoA show spatial overlap in metaphase suggesting they are active. Indeed, the DIAPH1 dependent γ-actin network contributes to metaphase cortical stiffness. However, during anaphase the β and γ-actin networks partition within the cell and are subject to independent and counter regulation. β-actin filament assembly is active at the equator, while γ-actin filament assembly is inhibited at the poles (Fig. 8). Little is known of the deactivation mechanisms of formins; however, our study provides insight into the principles of DIAPH1 deactivation that may be relevant to other formins. We demonstrate that DIAPH1 must relinquish its interactions with both RhoA and IQGAP to return to its inactive state. Thus, formin deactivation is not driven simply by the loss of RhoA-GTP binding in a process that would be a straightforward reversal of the activation mechanism[18], but rather exploits an independent deactivation pathway. We propose that CLIP170,

delivered to the cell cortex by astral microtubules during anaphase, displaces IQGAP1 from DIAPH1 and thus switches the formin OFF, leading to the depletion of γ-actin from the cell poles during cytokinesis. Little is known of the deactivation pathways of other formins, but the existence of a two stage formin deactivation mechanism would lend itself to a greater diversity of spatiotemporal inputs and a correspondingly increased subtlety of response, particularly if the in vivo efficiency of actin filament nucleation is tunable beyond the ON and OFF state, depending on the regulatory factors bound to formins.

Cytokinesis provides an ideal integrated model in which to dissect the coordinated and localized regulation of distinct cytoskeletal networks. The events at the cell poles are dependent upon the anaphase astral microtubules that emanate from the spindle poles to the cell cortex. In contrast, events at the equator are dependent upon the spindle midzone microtubules to

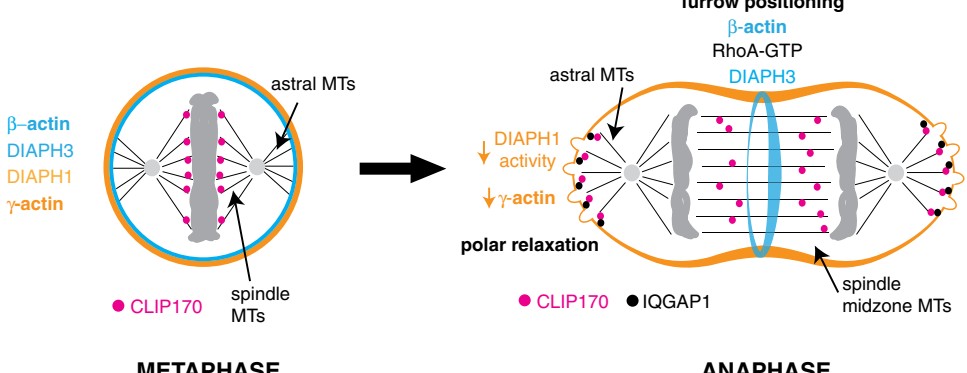

**Fig. 8 Schematic outlining of how the redistribution of different actin-isoform networks and their upstream regulators could be influenced by different anaphase microtubule structures in order to drive successful cytokinesis.** In metaphase different actin-isoform networks and their upstream regulators are equally distributed around the cortex. However, these networks are differentially redistributed around the cortex upon entry into anaphase. Spindle midzone microtubules (MTs) establish the spatial cues for active β-actin assembly and furrow ingression at the equator, while astral microtubules deactivate DIAPH1 at the cell pole, depleting the polar γ-actin network that relaxes the poles, allowing blebbing and the equalization of internal hydrostatic pressure. MTs microtubules.

position contractile ring assembly (Fig. 8). These observations suggest there are at least two distinct populations of microtubules in anaphase each with different properties that generate unique biochemical environments that drive localized cellular activities. One population of microtubules, the astral microtubules effect a reduction of polar cortical γ-actin, whilst another population of microtubules, the spindle midzone microtubules, effect positioning of an active Rho pathway to generate a β-actin network that forms the contractile ring and drives furrow ingression. What establishes and defines these populations remains to be determined, whether it be their different proximity to chromosomes and the Ran pathway or other signalling networks. These observations may explain why in monopolar cells undergoing cytokinesis the actin cytoskeleton becomes polarized[28,29]. Here the astral microtubules would be expected to have similar properties to the astral microtubules of wildtype cells as the cortex they contact has a reduced actin network. In contrast, those microtubules emanating from the chromosomes toward an actin rich cortex would be expected to have the characteristics of spindle midzone microtubules. The actions of these two microtubule arrays in wildtype cells coordinate the timing of chromosome segregation with the breaking of plasma membrane symmetry necessary for furrow ingression during anaphase. The different microtubule arrays amplify plasma membrane asymmetry through the counter regulation of distinct and independent actin-isoform networks: β-actin filament production at the equator drives contractile ring assembly, while localized polar depletion of the γ-actin network promotes blebbing. The combined effect is to drive efficient cytokinetic furrow ingression and successful cell division. The molecular pathways downstream of the spindle midzone microtubules have been studied extensively[1,2], but little is known of the astral microtubules dependent pathways that regulate actin dynamics at the cell poles. Recent work identified a role for the kinetochore-bound PP1-Sds22 phosphatase complex in dephosphorylating ERM proteins, causing a release of cortical actin[14]. If actin is to flow from the pole to the equator, releasing actin filaments from ERM proteins would be necessary; however, it remains unclear how kinetochore-bound factors would interact with membrane bound ones. In contrast, during *C. elegans* embryogenesis no role for the PP1-Sds22 complex was uncovered, rather a pathway dependent on TPX2-mediated activation of the Aurora-A kinase at the spindle poles was identified[30]. While these studies do not identify a direct link between the factors identified and the cortex of the

plasma membrane, they could be viewed as supporting a model whereby astral microtubules, perhaps through anaphase specific phospho-regulation pathways, regulate polar actin organization. Whether these mechanisms point toward anaphase specific regulation of astral microtubule cargos or changes in astral microtubule dynamics is not known.

Our study emphasizes that the actin cytoskeleton is both heterogeneous and dynamic, and that specialized actin-isoform networks perform specialized localized functions within the cell. During cytokinesis the differential regulation of independent β and γ-actin isoform networks enables the plasma membrane to be deformed in different ways at distinct locations to allow cytokinetic furrow ingression and the successful completion of cytokinesis. We have begun to define pathways upstream involved in the formation of distinct actin-isoform network. We envisage that distinct effectors will be recruited or modulated by the different actin-isoform networks to effect localized cellular activities. Defining these factors, their dynamics and interrelationships with the individual actin-isoform networks will be vital in understanding how actin networks drive different cellular and developmental processes.

## Methods

**cDNA cloning**. pGEX-6P-2-DIAPH1-CT(580–1272aa) and pET30a-DIAPH1-NT (1–575aa) plasmids were gifts from J. Copeland (Department of Cellular and Molecular Medicine, University of Ottawa). The IQGAP1 full-length cDNA was a synthetic gene corresponding to NP 003861 a gift from Drs Y. Tong and C. Arrowsmith (Structural Genomics Consortium, Toronto, Ontario, Canada). The CLIP170 full-length cDNA was a gift from Dr. John Brumell (Dept. Molecular Genetics, University of Toronto). pEGFP-C1 Lifeact-EGFP plasmid was gift from Dr C. McCulloch, University of Toronto.

To generate complementary DNAs (cDNAs) of full-length human DIAPH1 (1–1272aa), cDNAs fragments of human DIAPH1-NT (1–575aa), and DIAPH1-CT (580–1272aa) were amplified using the i-Max II DNA polymerase (Froggalab) using oligonucleotide primers. All primers are listed in Supplementary Table 1. The 3' oligo of DIAPH1-NT and the 5' oligo of DIAPH1-CT contain complimentary sequence. The PCR fragments were mixed and reamplified using 5' and 3' oligos of DIAPH1. cDNAs of DIAPH1-CT were generated and fused to phospholipase Cδ1-PH (PLCδ1-PH) using analogues strategy to generated cDNAs of PLCδ1-PH-DIAPH1-CT. Alternatively, PCR fragments of DIAPH1-CT were cloned using the TOPO Gateway system (Life Technologies) being first cloned into the entry plasmid vector pCR8/GW/TOPO, then moved into the destination vectors pKM596 (Addgene plasmid 8837) to generate MBP fusion proteins.

cDNAs of full-length human IQGAP1 (1–1657aa), CLIP170 (1–1320aa), or fragments of IQGAP1-DBR (1500–1657aa), CLIP170-NT (1–350aa), CLIP170-CT (500–1320aa) were amplified using the i-Max II DNA polymerase (Froggalab) using oligonucleotide primers listed in Supplementary Table 1. PCR fragments of IQGAP-DBR (1500–1657aa) were cloned into pDEST15 destination vector (Life

Technologies) using In-Fusion Cloning Kit (Clontech) to generate GST fusion proteins. PCR fragments of CLIP170-NT (1–350aa) or CLIP170-CT (500–1320aa) were cloned into pKM596 (Addgene plasmid 8837) using In-Fusion Cloning Kit (Clontech) to generate MBP fusion proteins.

**Protein expression and purification**. Recombinant proteins were purified from BL21 *Escherichia coli* cells transformed with plasmids containing His$_6$ or GST fusion proteins, or ER2523 E. *coli* (New England Biolabs) with plasmids containing MBP fusion proteins. Cells were grown in LB media at 37 °C to an optical density of 0.6 at A$_{600}$. Recombinant protein expression was induced by the addition of 1 mM isopropyl β-D-1-thiogalactopyranoside (IPTG) and further incubated at 16 °C overnight. Cells were harvested by centrifugation at 5000 × g for 20 min and stored in −80 °C.

To purify GST fusion proteins, BL21 E. *coli* cells were resuspended in 25 mM HEPES, pH 7.5, 250 mM NaCl, 100 mM KCl, 0.5 mM β- mercaptoethanol, 1 mM PMSF, and lysed by sonication. The lysates were cleared by centrifugation at 10,000 × g for 30 min at 4 °C and supernatant applied to glutathione beads (Invitrogen). The glutathione beads were washed with 10 column volumes of column buffer (CB) containing 25 mM HEPES, pH 7.5, 250 mM NaCl, 100 mM KCl, 0.5 mM β- mercaptoethanol, 1 mM PMSF, 0.1% (v/v) Triton X-100. The GST fusion proteins were eluted in CB containing 10 mM glutathione.

To purify MBP fusion proteins, ER2523 E. *coli* cells were harvested, lysed as described above. The lysates were cleared by centrifugation at 10,000 × g for 30 min at 4 °C then supernatant applied to amylose resin (New England Biolabs). The resin was washed with 10 column volumes of CB and the MBP fusion proteins were eluted in CB containing 10 mM maltose.

To purify His$_6$ fusion proteins, BL21 E. *coli* cells were resuspended in 25 mM HEPES, pH 7.5, 500 mM NaCl, 5% (v/v) glycerol, 5 mM imidazole, 0.5 mM β-mercaptoethanol, 1 mM PMSF, and lysed by sonication. The lysates were cleared by centrifugation at 10,000 × g for 30 min at 4 °C then supernatant applied to nickel-Sepharose beads (Amersham Biosciences). The beads were washed with 10 column volumes of His$_6$ column buffer (HCB) containing 25 mM HEPES, pH 7.5, 500 mM NaCl, 5% (v/v) glycerol, 5 mM imidazole, 0.5 mM β- mercaptoethanol, 1 mM PMSF, 0.1% (v/v) Triton X-100. The His$_6$ fusion proteins were eluted in HCB containing 500 mM imidazole. Eluted proteins were dialyzed against 10 mM HEPES, pH 7.6, 100 mM KCl, 2 mM MgCl$_2$, 50 mM sucrose for 16 h at 4 °C and concentrated using Millipore Ultrafree spin columns (Milipore, Ireland) with a 10-kDa cutoff. Proteins were then aliquoted, flash-frozen in N$_{2(l)}$, and stored at −80 °C.

**Nucleotide loading of proteins**. To generate GTP-loaded GST-RhoA for the in vitro binding assays, purified GST-RhoA fusion proteins were added to 25 mM EDTA, 1 mM DTT, and GTP or GDP added to a 100× molar excess of the proteins. The reactions were incubated on ice for 40 min before adding MgCl$_2$ to a final concentration of 50 mM. GTP-loaded proteins were then dialyzed, concentrated, and stored as described above.

**In vitro binding assays**. To determine if IQGAP1, or RhoA binding to DIAPH1-NT blocks the re-association of DIAPH1-NT and -CT, 0.05nmol His$_6$ -DIAPH1-NT was immobilized onto 25 μl nickel-Sepharose beads (Amersham Biosciences) in 100 μl of His$_6$ incubation buffer (HIB) containing 25 mM HEPES, pH 7.5, 120 mM NaCl, 1 mM EGTA, 0.3% (v/v) Triton X-100, 1 mM β- Mercaptoethanol and incubated for 1 h at 4 °C. The beads were washed in HIB (all subsequent His$_6$ protein washes were done in this buffer unless otherwise noted) and blocked with 3% (w/v) BSA for 20 min. The beads were then washed and incubated in the presence or absence of equimolar GST-IQGAP1 or GST-RhoA for 2 h at 4 °C, followed by a washing step. Equimolar MBP-DIAPH1-CT was then added to the beads and incubated for 2 h at 4 °C. Unbound protein was removed by washing the beads in HIB. The beads were reisolated by centrifugation at 13,000 × g for 5 min at 4 °C and boiled in SDS sample buffer then analyzed by western blotting using an anti-MBP monoclonal antibody (E8032, New England Biolabs, 1:2500 dilution) to detect co-purifying MBP-DIAPH1-CT.

To determine if CLIP170 displaces DIAPH1 from binding to IQGAP1, 0.05 nmol GST-IQGAP1-DBR was immobilized onto 25 μl GST beads (Invitrogen) in 100 μl of incubation buffer (IB) containing 25 mM HEPES, pH 7.5, 250 mM NaCl, 100 mM KCl, 0.5 mM β- mercaptoethanol, 1 mM PMSF, 0.1% (v/v) Triton X-100 and incubated for 1 h at 4 °C. The beads were washed in IB (all subsequent GST protein washes were done in this buffer unless otherwise noted) and blocked with 3% (w/v) BSA for 20 min. The beads were then washed and incubated with equimolar His$_6$-DIAPH1-NT for 1 h at 4 °C, followed by a washing step. Equimolar MBP-CLIP170-NT was then added to the beads and incubated for 1 h at 4 °C. Unbound protein was removed by washing the beads in IB. The beads were reisolated by centrifugation at 13,000 × g for 5 min and boiled in SDS sample buffer then analyzed by western blotting using an anti-His$_6$ polyclonal antibody (MP Biomedicals, 1:500 dilution) to detect co-purifying His$_6$-DIAPH1-NT, or an anti-MBP monoclonal antibody (E8032, New England Biolabs, 1:2500 dilution) to detect co-purifying MBP-CLIP170-NT. Alternatively, MBP-CLIP170-NT was added and incubated with the GST-IQGAP-CT immobilized on beads. The beads were then blocked and washed as described above. His$_6$-DIAPH1-CT was then added and incubated with the beads for 1 h at 4 °C. Unbound protein was removed

by washing the beads in IB. The beads were reisolated and analyzed as described above to detect co-purifying His$_6$-DIAPH1-NT or MBP-CLIP170-NT.

**Western blotting analysis**. Western blotting was performed according to standard procedures using following primary antibodies: anti-DIAPH1 antibody (No. 610848, BD Biosciences, 1:100 dilution), anti-IQGAP1 antibody (ab109292, Abcam, 1:250), anti-GFP antibody (sc-390394, Santa Cruz Biotechnology, 1:500 dilution), anti-MBP antibody (E8032, New England Biolabs, 1:2500 dilution), anti-6xHis antibody (MP Biomedicals, 1:500 dilution), a home-made anti-GST antibody, anti-RhoA antibody (sc-418, Santa Cruz Biotechnology, 1:200). The PVDF membrane of western blots was developed by chemiluminescent solution (Life Technologies) for 5 min at room temperature and visualized using a BioRad MP Imager (Bio-Rad, Canada).

**Generation of stable cell lines**. DIAPH1-I862A mutant was generated by PCR using oligonucleotides that encoded the mutation as listed in Supplementary Table 1. A full-length cDNA of wt-DIAPH1, a full-length cDNA of DIAPH1 including the I862A mutation, or a cDNA of PLCδ1-PH-DIAPH1-CT was amplified by PCR and cloned into a modified pcDNA 5 FRT/TO vector downstream of GFP using the In-Fusion Cloning Kit (Clontech). A full-length cDNA of IQGAP1 or a cDNA of Lifeact was cloned into the same vector using analogous strategy. Stable HeLa cell lines with regulated expression of GFP-transgenes were generated using the Flp-In system (Life Technologies) using HeLa cells that contained a single FRT site. The resulting cell lines were cultured in DMEM (Sigma) supplemented with 10% fetal bovine serum (Life Technologies), 5 μg/ml blasticidin (Bioshop), and 200 μg/ml hygromycin (Bioshop) in a 5% CO$_2$ atmosphere at 37 °C. GFP fusion protein expression was induced by incubating stable HeLa cell lines with 1 μg/ml Dox for 24 h and expression of the GFP fusion protein detected by western blotting with an anti-DIAPH1 antibody (No. 610848, BD Biosciences, 1:100 dilution), or an anti-IQGAP1 antibody (ab109292, Abcam, 1:250 dilution), or an anti-GFP antibody (sc-390394, Santa Cruz Biotechnology, 1:500 dilution).

**siRNA knockdown assays**. HeLa cells (obtained from Dr. Laurence Pelletier, University of Toronto) were transfected with 40 nM double-strand DIAPH1, DIAPH3 or IQGAP1 siRNA using Lipofectamine 2000 Reagent (Invitrogen) or a negative control siRNA NC1 (IDT). For rescue experiments, 16–24 h after siRNA treatment, cells were treated with 1 μg/ml Dox to induce the expression of GFP-DIAPH1 or GFP-IQGAP1 transgene. siRNA were obtained from Integrated DNA technologies. The siRNA duplexes used in the assays are listed in Supplementary Table 2.

**In situ proximity ligation assays (PLA)**. PLA assays were performed using Duolink® In Situ Red Kit Mouse/Rabbit (Sigma) following the manufacturer's protocol. Briefly, to detect DIAPH1-IQGAP1 interaction, HeLa cells were fixed with methanol at −20 °C for 15 min and blocked by Duolink® Blocking Solution for 60 min at 37 °C. A mouse anti-DIAPH1 antibody (No. 610848, BD Transduction Laboratories) or a rabbit anti-IQGAP1 antibody (ab109292, Abcam) was diluted in Duolink® Antibody Diluent in a 1:100 or 1:250 ratio, respectively, and incubated with the cells for 60 min at room temperature. Alternatively, the DIAPH1 antibody described above and a Rabbit anti-CLIP170 (ab106524, Abcam) was diluted in Duolink® Antibody Diluent in a 1:100 or 1:50 ratio, respectively, to detect DIAPH1-CLIP170 interaction using the same methods described above. The cells were then washed twice with Duolink® Anti-Mouse PLUS and Anti-Rabbit MINUS probes for 60 min at 37 °C, followed by washing the cells twice in Wash Buffer A. The cells were then incubated with Duolink® Ligase diluted in ligation buffer in a 1:40 ratio for 60 min at 37 °C, followed by washing the cells twice in Wash Buffer A. The Duolink® Polymerase diluted in amplification buffer in a 1:80 ratio was added to the cells and incubated for 100 min at 37 °C, followed by washing the cells 2 × 10 min in Wash Buffer B at room temperature. The cells were then stained with 4′, 6-diamidino-2-phenylindole (DAPI) to visualize DNA.

To detect IQGAP1-CLIP170 interaction, HeLa cells were fixed with 3.7% PFA for 15 min at room temperature, permeabilized for 10 min with 0.2% Triton-X-100 in PBS and blocked using the same methods described above. A mouse anti-CLIP170 antibody (sc-28325, Santa Cruz Biotechnology) or a rabbit anti-IQGAP1 antibody (ab109292, Abcam) was diluted in Duolink® Antibody Diluent in a 1:500 or 1:250 ratio, respectively, and incubated with the cells for 60 min at room temperature. The PLA assays were then performed using the same methods described above.

**Inhibition of centrosome duplication**. HeLa cells were cultured in a 10-cm dish in DMEM (Sigma) supplemented with 10% fetal bovine serum (Life Technologies) in a 5% CO$_2$ atmosphere at 37 °C until ~60% confluency was reached. Mitotic cells were harvested by tapping gently on the side of the dish in 2 ml of media, with floating cells (mitotic or early G1 cells) collected and reseeded in a new dish. Centrinone B was added to the reseeded cells after 2 h to a final concentration of 100 nM and incubated for another 48 h. The cells were then fixed and analyzed by immunofluorescence staining.

**Immunofluorescence and microscopy analysis**. To visualize HeLa cells expressing GFP fusion proteins, cells were fixed with 3.7% PFA for 15 min at room temperature, permeabilized by 0.2% Triton X-100 in PBS then stained with 4′, 6-diamidino-2-phenylindole (DAPI) to visualize DNA. To visualize cellular β- or γ-actin structures, cells were fixed with prewarmed 3.7% PFA for 30 min at 37 °C. Cells were washed three times with PBS, then post-fixed with methanol at −20 °C for 15 min. Cells were stained by a mouse anti-β actin or a mouse anti-γ antibodies (Bio-Rad, 1:400) overnight at 4 °C, followed by washing the cells three times in PBS. Secondary goat anti-mouse antibodies conjugated to Alexa 594 (Invitrogen, 1:1000) were used to visualize the cellular β- or γ-actin structures. Coverslips were mounted on glass slides using Mowiol (Polyvinyl alcohol 4-88, Fluka).

To visualize cellular localization of DIAPH1 or Aurora-A, cells were fixed with methanol at −20 °C for 15 min and blocked by 3% BSA in PBS for 1 h at room temperature. Cells were stained by an anti-DIAPH1 antibody (No. 610848, BD Transduction Laboratories, 1:100), or an anti-Aurora-A antibody (ab115883, Abcam, 1:600) overnight at 4 °C. Subsequently cells were washed three times in PBS. A secondary donkey anti-mouse antibody conjugated to Alexa 594 (against anti-DIAPH1 antibody) (Life Technologies, 1:100 dilution), or a donkey anti-goat antibody conjugated to Alexa-488 (against anti-Aurora-A antibody) (Life Technologies, 1:1000 dilution) was used to visualize DIAPH1 or Aurora-A localization, respectively. Cells were then stained with 4′,6-diamidino-2-phenylindole (DAPI) to visualize DNA.

To visualize cellular localization of RhoA, cells were immersed in ice-cold 10% trichloroacetic acid (TCA) dissolved in distilled water for 15 min on ice. Cells were then washed with 30 mM glycine in PBS three times. Cells were permeabilized for 10 min with 0.1% Triton X-100 in PBS then blocked by 3% BSA in PBS for 1 h at room temperature. Cells were stained by an anti-RhoA antibody (sc-418, Santa Cruz Biotechnology, 1:500). Subsequently cells were washed three times in PBS; a secondary goat anti-Mouse antibody conjugated to Alexa 594 (against anti-RhoA antibody) was used to visualize RhoA localization.

To visualize cellular localization of IQGAP1, cells were fixed with 3.7% PFA at room temperature for 15 min, permeabilized for 10 min with 0.1% Triton X-100 in PBS then blocked by 3% BSA in PBS for 1 h at room temperature. Cells were stained by an anti-IQGAP1 antibody (ab109292, Abcam, 1:250). Subsequently cells were washed three times in PBS; a secondary goat anti-Rabbit antibody conjugated to Alexa 594 (against anti-IQGAP1 antibody) was used to visualize IQGAP1 localization. To visualize cellular localization of microtubules, cells were fixed and blocked using the same methods above but stained by an anti-α-tubulin antibody (ab7291, Abcam, 1:1000) followed by a secondary goat anti-mouse antibody conjugated to Alexa-488 (Life Technologies, 1:1000). Cells were then stained with 4′,6-diamidino-2-phenylindole (DAPI) to visualize DNA.

Coverslips were mounted on glass slides using Mowiol (Polyvinyl alcohol 4-88, Fluka). Cells were visualized using a Perkin Elmer UltraView spinning disk confocal scanner mounted on a Nikon TE2000-E with a ×60/1.4 NA oil-immersion objective lens and 1.515 immersion oil at room temperature. Images were acquired using METAMORPH software (Molecular Devices) driving an electron multiplying charge-coupled device (CCD) camera (ImagEM, Hammamatsu) Z sections (0.2 μm apart) were acquired to produce a stack that was then imported into AUTO-QUANT X2 (Media Cybernetics) for deconvolution (10 iterations). Maximum projections and cross sections were performed using METAMORPH. Images were overlaid in PHOTOSHOP (Adobe), involving adjustments in brightness and contrast of images.

To perform live-cell imaging, cells cultured on circular glass coverslips, thickness no. 1, diameter 25 mm (Fisher Scientific) were treated as indicated and mounted in a heated chamber containing air–5% CO₂ atmosphere at 37 °C (Live Cell Instrument Systems) in dye-free DMEM with 10% FBS (Invitrogen) mounted on a Nikon TE2000 inverted microscope equipped with a spinning disc confocal scanning head driven by Metamorph software as described above. Time-lapse video microscopy was used to follow cells with a stack of images (z-step 2 μm) taken every 90 s using a ×60/1.0 NA oil-immersion objective lens and 1.515 immersion oil (Nikon).

**Atomic force microscopy (AFM) analysis**. Cell stiffness was evaluated using standard AFM-based nano-indentation measurements performed with a spherical tip (diameter −20 μm) attached to a silicon nitride cantilever (elastic constant 0.038 N/m and nominal resonance frequency of 10 kHz, Novascan) as previously described, with minor modifications[31,32]. Briefly, nano-indentation measurements were performed with the spherical tip applied to the cell surface in the region of the nucleus. Measurements were performed at room temperature in the liquid phase using a commercial Bruker Resolve (Bruker, USA) AFM combined with an inverted optical microscope. Cantilever deflection and z-piezo movements were detected at each indentation step to produce a force-displacement curve by knowing the cantilever spring constant[33]. The Young's modulus was calculated from the force-displacement curves, at an indentation around 500 nm, employing the Hertz model, according to standard techniques[34–36]. This prevented plasma membrane damage and allowed us to sample the mechanical properties of the plasma membrane and underlying actin cytoskeleton[34].

The Hertz model describes the elastic deformation of two spheres, where the relation between the applied force (F) and the resulting indentation is:

$$F = \frac{4ER^{\frac{1}{2}}}{3} \frac{\delta^{\frac{3}{2}}}{(1-\mu^2)}$$

where E is Young's modulus, μ is the Poisson's ratio of the indented material, and R is the radius of the rigid indenter. For our calculations, we used a μ of 0.5, the standard value for living cells[32].

**Biolayer-interferometry (BLI) assays**. Dissociation constants between different proteins were determined by the Octet RED96 system (FortéBio), which measures association onto and dissociation from a sensor surface using Biolayer interferometry (BLI) as previously described[17,33]. Briefly, purified GST-tagged ligands (GST-IQGAP1-DBR or GST-DIAPH1-CT) were equilibrated into kinetics buffer (KB), which contains phosphate-buffered saline (PBS), 0.002% Tween-20, and 0.1 mg/ml BSA, to a concentration of 25 μg/ml. The GST biosensors (FortéBio) were first equilibrated in KB for 60 s and then followed by incubation with GST ligands in KB for 300 s. Sensors were then rinsed in KB for 250 s to obtain a baseline of the levels of GST ligands loading onto the biosensors. Binding assays were then performed in a series of increased concentration of analyte (MBP-CLIP170-NT or MBP-CLIP170-CT) from 0.5 nM to 5 μM, from the lowest concentration to the highest. Each binding sequence started with a baseline incubation in KB (150 s), followed by the association with the analyte (300 s), the dissociation in KB (250 s), and a regeneration step (200 s) where sensors were rinsed in regeneration buffer (100 mM sodium citrate pH = 4.5, 50 mM EDTA, and 150 mM NaCl), to remove analyte still bound to GST ligands. Reference sensors that were loaded with GST ligands but only assayed in a series of pure KB, or a series of increasing concentration of MBP were also measured as controls. All incubation steps were performed at 30 °C with a shaking speed of the plates at 1000 rpm. Data analysis was performed using Octet Software (FortéBio) and GraphPad Prism (8.2.0). The control signals measured by the reference sensors were subtracted from the signals measured by analyte-bound sensors. Binding data were analyzed by Octet software assuming one site-specific binding to generate a saturation binding curve where the dissociation constant ($K_d$) was derived. Each binding sequence was repeated five times. The resulted $K_d$ values and averages were plotted in GraphPad Prism (8.2.0), with error bars representing ±S.D.

**Quantification and statistical analysis**. The quantification of the numbers of multinucleate cells was performed using GraphPad Prism (v8.2.0). Each data point represents the percentage of multinucleate cells in a population of 30 cells within an optic field of the ×60/1.4 NA oil-immersion objective lens. The quantification of the numbers of cells with DIAPH1 or IQGAP1 at the cortex was performed using the same method described above, where each data point represents the percentage of cells with cortical protein staining in a population of 30 cells within an optic field. At least 20 random fields from three independent experiments were analyzed. Mean and ±s.d values were calculated in each group.

The number of blebs around the cortex, or length of long axis of the cell was determined in individual cell and at least 100 cells from three independent experiments were analyzed. Mean value and 25–75 percentile were calculated in each group. The number of PLA signals around the cortex or ratios of PLA signals between the two poles, the ratio of protein fluorescence intensity, or blebs between two poles were determined in individual cell and at least 50 cells from three independent experiments were analyzed. Mean and ±SD values were calculated in each group. Unless otherwise noted, the nonparametric unpaired two-tailed Mann-Whitney t-test was used to determine p-values in all statistical analysis in this study.

To quantify the rate of furrow ingression, the depths on both sides of the cytokinetic furrow ($d_1$, $d_2$), and the width of the collar (C) were first measured (Supplementary Fig. 2). The average depth of furrow ingression (FI) is defined as $(d_1 + d_2)/2$. The relative depth of furrow ingression (RFI), which can be compared among different cells, is defined as FI/C. The bigger the RFI, the greater extent of furrow ingression. A RFI value around 3.0 indicates the end of anaphase B where the collar is still clear and observable. The average rate of furrow ingression is calculated as the slope by fitting RFI (≤3.0) as a linear function of time (t). Calculated rates are normalized to control siRNA treated group (set to 1.0). A total of five cells in each live-cell imaging groups were analyzed (n = 5). Mean and ±s.d. values were calculated in each group. Data and graphs were plotted in GraphPad (8.2.0).

Microscopic analysis for each experiment in this study (Fig. 1a, c, d; Fig. 3a, c; Fig. 4a; Fig. 5a; Fig. 7a, b, d; Supplementary Fig. 1c, d, e; Supplementary Fig. 5a, b, c, d, e) was repeated three times independently with similar results. Micrographs presented in the figures were representative of three independent experiments. Western blotting analysis for each experiment in this study (Fig. 5b; Fig. 6a; Supplementary Fig. 1b, e; Supplementary Fig. 5f; Supplementary Fig. 6; Supplementary Fig. 7; Supplementary Fig. 8) was repeated three times independently with similar results. Immunoblots presented in the figures were representative of three independent experiments.

**Reporting summary**. Further information on research design is available in the Nature Research Reporting Summary linked to this article.

## Data availability
All data are available in the main text or the supplementary materials. Any plasmids and cell lines generated during and/or analysed during the current study are available from the corresponding author on reasonable request.

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

## Acknowledgements

We thank Dr C. McCulloch for reading the manuscript, C. Yip for use of the AFM and Drs J. Loncarek and L. Pelletier for advice on the use of Centrinone B. A.W. is supported by Canadian Institutes of Health Research (CIHR) Grant PJT 148575, T.F.M. is supported by infrastructure support provided by Canadian Foundation for Innovation and operating support from the Natural Sciences and Engineering Research Council of Canada (RGPIN-2018-06546). T.F.M. is a Tier II CRC in the Structural Biology of Membrane Proteins.

## Author contributions

A.C. performed all experiments except immunoprecipitations that were performed by T.C.P., the AFM which was performed by L.U.S. and D.A.Y. and BLI performed by A.C. and T.F.M. A.C., B.D.L., and A.W. conceived the experiments and wrote the manuscript that was edited by all authors.

## Competing interests

The authors declare no competing interests.
