## [Peer Review File · Nature Communications]

REVIEWER COMMENTS

Reviewer #1 (Remarks to the Author):

This paper by Chen et al., takes a fresh look at the old and important problem of how polar relaxation might contribute to cell division in human cells. Recently, after most effort focused on mid-zone furrowing, several groups have revisited cortical relaxation at cell poles and have identified potential roles for: i) Myosin flow away from the pole, ii) Inhibition of the cortex via Ran-GTP and by iii) PP1-Sds22. However, as the authors state in their manuscript, the extent to which MT-based or chromatin-based signals operate to induce cortical relaxation at cell poles in human cells remains unclear. Thus, much remains to be discovered. While it is hard to make progress because of the challenges of imaging cells as they change shape and rapidly transit from metaphase into interphase, Chen et al. rise to the challenge.

Overall their paper is very well written and the images are clear, and data, for the most part, has been quantified. Thus, the paper has many strengths. I especially like the analysis of cells with one centrosome. The polarization of δ actin under these conditions is striking.

There is one major weakness, however. While the first part of the paper involves the study of local roles for β and γ actin / DIAPH1 and DIAPH3 using cell biology, and a nice trick in which they study cells that contain a single spindle pole, the latter part focuses on the biochemistry of CLIP170-IQGAP-DIAPH1. In this part the authors aim to explore a hypothesis about the regulation of DIAPH1 inactivation. While this is important and may explain much of what they see in cells, there is a disjoin. Some elements are subjected to a cell biological analysis some to a biochemical analysis. The paper does not explore:

- i) How changes to CLIP170 impact polar relaxation.
- ii) The extent to which CLIP170 co-localizes with DIAPH3 at the cell equator.
- iii) Conversely, the biochemistry does not explore whether DIAPH3 might be a target of CLIP170.

In summary, a paper that focused on the cell biology would add much to the field.

If the biochemistry and cell biology are to be combined more work is required to ensure that alternative hypothesis are tested.

Main comments:

1. The analysis of anaphase elongation should be presented by showing the change in cell length from metaphase, not as a simple length measurement (which depends on timing).
2. Throughout the paper the images tend to focus on cells with a large furrow. However, furrowing itself changes things. Where possible, it would be good to show earlier stages of anaphase before furrowing. For example in Fig 1a and 1c. This is important because the authors propose that the symmetry breaking is induced by astral MT-based CLIP170. To show that this is the case, it is important that images show protein localizations (co-localizations (CLIP170/IQGAP1/DIAPH1) prior to furrowing. In fact, in 7b IQGAP – CLIP appears uniform at stage where symmetry breaking is occurring.
3. Please show test for specificity of beta and gamma actin stainings (using specific siRNAs or over-expression), and quantify the cortical profile of beta and gamma actin – to make clear what % is at

poles and centres. It is clear from images that both are present in both places, even if the author's qualitative statements are correct.

4. It would be good to have mechanical data for beta/gamma actin and DIAPH3 siRNA experiments in metaphase/pro-metaphase. This would help to show how they contribute to cortical stiffness. If the authors are unable to provide these data (e.g. because of the lockdown), they should make the caveats clear.

5. It is important to show the DIAPH3/DIAPH1 double KD. Following the logic of the paper this should suppress the furrow defects seen in DIAPH3 siRNA. However, it is possible that DIAPH1 substitutes for DIAPH3 in the siRNA condition. Please also show DIAPH1 and DIAPH3 localizations in the absence of the other.

6. Fig3. In 3a please show how measurements were made using boxes.

In 3c – why is there so much anaphase elongation with DIAPH3 siRNA and Pm DIAPH1?

This suggests that there is a second cue that polarizes the cortex.

If possible, please quantify the dynamics of the elongation at anaphase under these conditions.

7. The authors state "As astral microtubules are required for polar relaxation during anaphase".

Please show cells with no centrosomes and present the extent of anaphase elongation to make the extent of the defect clear in their system. It is very unlikely to be as black and white as this.

8. In fig 3c, both DIAPH3 and DIAPH1 appear to have blebs at metaphase – please quantify.

9. If the author's model is correct, why does the cortex clear on the side far from astral MTs in monopolar human cells forced to leave mitosis? This is an important issue to address and to discuss in the paper. It would be great if the authors could reverse this, e.g. in a DIAPH3 KD.

Specific comments:

1. While the focus is on human cells, it would be good to comment on other systems (flies and worms). How would findings made studying β and γ actin / DIAPH1 and DIAPH3 isoforms play out in organisms where cytokinesis is likely to be mediated by a single actin and Dia homolog?

2. There is no mention of Arp2/3. This has been shown to play an important role in the HeLa cortex and in anaphase in many systems. How much of the γ actin is nucleated downstream of Arp2/3?

3. The discussion on blebs ignores the fact that they are sites of actin and Myosin II recruitment. How do the authors imagine this happens at cell poles? Note, a loss of membrane blebs can occur with a uniform gain of cortical tension as well as a loss. Blebs happen when there is an imbalance of forces.

4. The authors state "As anaphase proceeded, we observed that the Lifeact-GFP signal decreased at the cell poles (Fig. 3a) due to the cortical flow of actin from the poles to the equator". The authors shouldn't assume this is true without measuring it. Temper the language here, i.e. remove 'due to'.

5. The authors state "Surprisingly, the loss of the b-actin contractile ring upon depletion of DIAPH3 or anillin, an enhancer of DIAPH3, also allowed a rudimentary, although unstable furrow to ingress

(Fig. 3b, c, Supplementary Fig. 2, Supplementary video 4) 9,12,18." This is likely due to DIAPH1. Test and/or discuss.

6. the authors state: "As astral microtubules are required for polar relaxation during anaphase..." Again, show this or temper the language.

7. It would be good to add a label 5a Rho (which Ab?)

Reviewer #2 (Remarks to the Author):

The manuscript authored by Chen et al. describes the role of β - and γ -actin in cytokinesis. By using time-lapse fluorescence observation of actins, formins and GAPs, they concluded that β -actin network is formed at the cleavage furrow mainly by DIAPH3, whereas γ -actin localized at the cortex except for cell poles, which is caused by deactivation of DIAPH1. They also utilized atomic force microscopy (AFM) to measure the stiffness of the cell cortex during mitosis. Mechanical characterization of mitotic cells by AFM has been conducted by a number of previous studies, and a well-established approach. However, the measurement and the analysis requires highly careful considerations depending on what is to be measured. Unfortunately, the experimental procedures of the AFM experiments in this study were not well-described in the manuscript, and therefore difficult to understand. The followings are the points that the authors should clarify.

i) In figure 2, the authors tried to compare "membrane tension" in normal cells and formin-knock-down cells to explain delocalized and increased membrane blebbing they observed in figure 1. I agree that the membrane blebbing largely depends on a local membrane tension, which is determined not only by osmotic pressure but also interaction between cortical skeleton and the lipid bilayer. Therefore, measuring membrane tension in various knock-down cells may provide a useful information. However, on the other hand, the experimental approach that they utilized to measure the "membrane tension" seems inappropriate. The authors claimed they measured "membrane tension" by recording and analyzing the force-indentation curve of AFM cantilever. However, the force curve obtained by pushing the cantilever against the cell surface contains the stiffness of both plasma membrane and cortical skeleton. Since the cortical skeleton (cortical actin) is much stiffer than the plasma membrane, the Young's modulus obtained from the force curve mostly indicates the stiffness (elasticity) of the cortex and not that of the plasma membrane. Since the authors did not describe the detail procedure of the force measurement (loading rate, indentation depth, etc.) it is hard to understand how they extracted the information of membrane tension and exclude the stiffness of the cortex. Or, do they want to measure the elasticity of the cortical skeleton? If so, how they link it to the blebbing? The authors have to describe what they want to measure (membrane tension or elasticity of the cortex) by AFM, and show the detail procedure to extract the information they want from the obtained force curve.

ii) The authors described that they obtained the force-indentation curve from metaphase cells. I wonder why they used metaphase cells and not the telophase and cells in cytokinesis, which is the most important steps of actin and formin function. They showed different distributions of β - and γ -actins, as well as DIAPH1 and 3, during cytokinesis (figure 1), but not in metaphase. It has been known that the properties of the cell cortex drastically changes upon the transition from metaphase

to anaphase. So, I wonder why they measured the elasticity of metaphase cells, and not in the following period, which is highly confusing. The authors should clarify this point. If possible, they should perform the same type of force measurement with telophase cells or cells in cytokinesis.

iii) As mentioned above, the description of the experimental procedure of AFM measurement is insufficient. They just cited a paper (Santoro R., et al.) in which the elasticity of interphase cells were measured by AFM. However, since the mechanical properties of the adherent cells in interphase is different from that of mitotic cells, they have to optimize the procedures of the measurement and analysis. Especially, if they want to measure the “membrane tension”, the indentation depth and the way of curve fitting is critical (in the cited paper, indentation depth was 500 nm, but this is too far to measure the properties of the plasma membrane). I strongly recommend the authors should show raw data of force curve and explain how they fit it with Hertzian model. It may be necessary to show representative force curves from individual knock-down cells as supplementary.

Reviewer #3 (Remarks to the Author):

In this remarkable and excellent study, the authors provide multiple unique and fundamental novel insights into how γ -actin at the poles of anaphase cells is regulated to permit efficient cytokinesis. They first establish the differential distribution of β - and γ -actin, at the furrow and along the cortex respectively during anaphase, and the related formins DIAPH1 and DIAPH3. Disruption of DIAPH1 or its activator IQGAP1 induce the loss of cortical γ -actin, blebbing around the anaphase cells and increase in multi-nucleated cells indicative of cytokinesis failure. Cells lacking function DIAPH1 are much less stiff as assessed by AFM, as expected. Next, they show that appropriate furrow ingression requires actin assembly by both DIAPH1 and DIAPH3. In a nice experiment following treatment with centrinone B to limit Aurora-A to just one spindle pole body and so limit astral microtubules to that pole, they show that DIAPH1 and γ -actin removal at the pole requires astral microtubules. Next, they explore the mechanism of DIAPH1 inhibition at the pole. These formins are regulated by an N-/C-terminal inhibitory interaction. RhoA or IQGAP binding to the N-terminal region of DIAPH1 can abolish its ability to bind the catalytic C-terminal region. Based on studies that showed the activator IQGAP1 can bind the microtubule associated protein CLIP170, they find that CLIP170 can sequester IQGAP1 from DIAPH1 and thereby remove the activator. Finally, using proximity ligation assays they show that in metaphase, IQGAP1 associated with DIAPH1 at the cell cortex and then with CLIP170 in anaphase, indicating that the delivery of CLIP170 by astral microtubules locally deactivates DIAPH1 at the poles, and this contributes to the formation of the cleavage furrow.

Building on their 2017 Nat. Comm. paper, this provides many notable advances to very long-standing issues: the clearest functional distinction between β - and γ -actin; two related formins that drive assembly of selected actin isoforms; regulation of both cortical γ -actin and cytokinetic β -actin are needed for furrow ingression; that CLIP170 can remove the formin activator IQGAP1; this regulation is mediated in a temporal and spatial manner by astral microtubules. This is one of the best papers I have read in a long time - superior in results and message to many in the major CNS journals.

My only comment relates to the distribution of γ -actin. Wouldn't one expect γ -actin to be diminished from the poles (as diagrammed in Fig. 8) at late anaphase in Fig. 1A and Fig. 4a control and as reported by others (eg. Rodrigues et al. Nature 524, 489)? In the centrinone B experiment, it seems it is diminished where there are astral microtubules. A statement about this would clarify the

issue.

Response to reviewers

We thank each of the reviewers for their constructive analysis of our study. We have made modifications to the manuscript to accommodate their requests where possible and have included data below to clarify findings and to explain why certain experiments were not technically possible. We note that while the modifications to our manuscript have strengthened our study, none of our original conclusions have changed.

A detailed response to each reviewer comments is below.

Response to reviewer 1

Reviewer #1 (Remarks to the Author):

This paper by Chen et al., takes a fresh look at the old and important problem of how polar relaxation might contribute to cell division in human cells. Recently, after most effort focused on mid-zone furrowing, several groups have revisited cortical relaxation at cell poles and have identified potential roles for: i) Myosin flow away from the pole, ii) Inhibition of the cortex via Ran-GTP and by iii) PP1-Sds22. However, as the authors state in their manuscript, the extent to which MT-based or chromatin-based signals operate to induce cortical relaxation at cell poles in human cells remains unclear. Thus, much remains to be discovered. While it is hard to make progress because of the challenges of imaging cells as they change shape and rapidly transit from metaphase into interphase, Chen et al. rise to the challenge.

Overall their paper is very well written and the images are clear, and data, for the most part, has been quantified. Thus, the paper has many strengths. I especially like the analysis of cells with one centrosome. The polarization of δ actin under these conditions is striking.

There is one major weakness, however. While the first part of the paper involves the study of local roles for β and γ actin / DIAPH1 and DIAPH3 using cell biology, and a nice trick in which they study cells that contain a single spindle pole, the latter part focuses on the biochemistry of CLIP170-IQGAP-DIAPH1. In this part the authors aim to explore a hypothesis about the regulation of DIAPH1 inactivation. While this is important and may explain much of what they see in cells, there is a disjoin. Some elements are subjected to a cell biological analysis some to a biochemical analysis

The paper does not explore:

i) How changes to CLIP170 impact polar relaxation.

Response: The reviewer raises an important point that we tried hard to address prior to our initial submission. However despite our best efforts, we observed that disruption of CLIP170, for example by siRNA induced depletion, had a severe effect on cell fitness. There was a significant reduction in viable cells upon CLIP170 siRNA treatment compared to control siRNA treatment (See reviewer Fig 1a below). Of the cells remaining, almost none were in anaphase or telophase, the stages of the cell cycle needed to assess CLIP170's role in cytokinesis. There was a slight increase in cells in prometaphase and metaphase, suggesting that the CLIP170 depleted cells became stalled at these stages of mitosis (See reviewer Fig 1b below). These observations are consistent with those previously reported by Wieland *et al* (MCB 2004 24:6620-6630). Our

observations in fixed cells were supported by live imaging analysis, where cells failed to progress from metaphase to anaphase. This failure of cells to progress from metaphase to anaphase is not surprising as Tanenbaum *et al* (EMBO J. 2006 25:45-57, work corroborated by others) discovered that CLIP170 is required for the attachment of microtubules to kinetochores. Consequently, cells depleted of CLIP170 failed to congress their chromosomes to the metaphase plate which would have the consequence of preventing a cell progressing into anaphase.

While this aspect of CLIP170 biology does not allow us to analyse its role in anaphase we do visualize within a cell using a proximity ligation assay that CLIP170 interacts with IQGAP1 only after metaphase.

ii) *The extent to which CLIP170 co-localizes with DIAPH3 at the cell equator.*

Response: We have added micrographs of CLIP170 and DIAPH3 localization from metaphase through anaphase that reveal no significant co-localization of CLIP170 and DIAPH3 (supplementary Fig 5). Indeed, in the pre-furrowing cells, there is a no significant enrichment of CLIP170 at the equator nor do we detect no CLIP170-DIAPH3 PLA foci in metaphase or anaphase. In addition, no interaction between anillin (a DIAPH3 activity enhancer) and CLIP170 was observed in PLA assays. These data along with our interaction analysis (see below) suggests CLIP170 exerts little or no influence over equatorial events, in particular the DIAPH3 pathway.

iii) Conversely, the biochemistry does not explore whether DIAPH3 might be a target of CLIP170.

Response: We have added immunoprecipitation data from HeLa cell lysates that reveal no co-immunoprecipitation of CLIP170 with GFP-DIAPH3 nor GFP-anillin (supplementary Fig 5). In contrast, CLIP170 coimmunoprecipitates with GFP-IQGAP1, as expected from our biochemistry (Fig 5). GFP-DIAPH3 does co-immunoprecipitate with anillin, but GFP-anillin does co-immunoprecipitate with CD2AP as expected from our previously published biochemical studies (Chen *et al* 2015, 2017). Our previously published studies (Chen *et al* Nat. Comm., 2017 and Chen *et al* J.B.C., 2020) demonstrated that while the activation mechanism of DIAPH1 and DIAPH3 follow the same principles, i.e. RhoA-GTP binds first followed by an enhancer, the enhancers differ between the two different formin pathways and show no-cross interactions. These results combined with our current observations, both *in vitro* biochemical and *in cellula* localization data, demonstrate that DIAPH1 and 3 operate in different pathways.

In summary, a paper that focused on the cell biology would add much to the field. If the biochemistry and cell biology are to be combined more work is required to ensure that alternative hypothesis are tested.

Main comments:

1. *The analysis of anaphase elongation should be presented by showing the change in cell length from metaphase, not as a simple length measurement (which depends on timing).*

Response: We include an analysis of anaphase cell elongation from our live imaging that analysis reveals changes in cell elongation rates. The text has been amended to include this point. In our original submission, we made use of the far greater cell numbers derived from our fixed cell analysis (vs the smaller number from live cell analysis) and measured the length of the cells as both fixed chromosomal distances and fixed furrow ingression depth. We have moved this data to the supplementary section. The combination of fixed cell analysis and live cell analysis strengthens our observation that the process of cell elongation is disrupted.

2. *Throughout the paper the images tend to focus on cells with a large furrow. However, furrowing itself changes things. Where possible, it would be good to show earlier stages of anaphase before furrowing. For example in Fig 1a and 1c. This is important because the authors propose that the symmetry breaking is induced by astral MT-based CLIP170. To show that this is the case, it is important that images show protein localizations (co-localizations (CLIP170/IQGAP1/DIAPH1) prior to furrowing. In fact, in 7b IQGAP – CLIP appears uniform at stage where symmetry breaking is occurring.*

Response: We have added images to Fig 1a of the actin isoforms, formins and IQGAP1 at metaphase and early anaphase just prior to furrowing. These images clearly demonstrate that even prior to furrowing the actin isoforms and formins begin to assume asymmetric distributions along the plasma membrane. For Fig 1c we have added the images of earlier stages into a supplementary figure, so as not to make Figure 1 so enormous that individual panels would not be readable. These images are all consistent with those in Fig 1a.

3. Please show test for specificity of beta and gamma actin stainings (using specific siRNAs or over-expression), and quantify the cortical profile of beta and gamma actin – to make clear what % is at poles and centres. It is clear from images that both are present in both places, even if the author's qualitative statements are correct.

Response: The specificity of the actin isoform antibodies was extensively characterized in their initial description by the Chaponnier Lab (Dugina *et al* 2009) and by others including our previous study, Chen *et al* 2017 and in our current study that shows when different formins are depleted different subsets of the actin cytoskeleton are depleted. Previous analysis included the demonstration of antibody specificity in Western blot experiments of 2D gels where the actin isoforms were separated from one another (Dugina *et al* 2009 Fig 1 and Chen *et al* 2017 Sup Fig 4) and immunofluorescence-based assays. To complement these data, we have included reviewer Fig 2 below, where β or γ -actin are depleted from cells by siRNA. As the figure demonstrates, depletion of β -actin results in no detectable filaments with the β -actin antibody whilst in the same cell filamentous staining is seen with the γ -actin antibody. The converse is true in cells where γ -actin is depleted by specific siRNA.

To further demonstrate that β -actin is enriched in the equatorial region of the anaphase cell where furrowing occurs and that γ -actin is more uniformly distributed around the plasma membrane we have added quantification of the respective antibody staining into Supplementary Fig 1a.

4. It would be good to have mechanical data for beta/gamma actin and DIAPH3 siRNA experiments in metaphase/pro-metaphase. This would help to show how they contribute to cortical stiffness. If the authors are unable to provide these data (e.g. because of the lockdown), they should make the caveats clear.

Response: The AFM data was collected from cells in metaphase. We have further clarified this in the text. Our goal was to collect AFM data during anaphase, however due to the large increase in dramatic blebbing events in some conditions during anaphase (e.g. DIAPH1 depletion) it was not possible to collect such data. Therefore, to collect a complete set of comparable data, we restricted our analysis to metaphase.

5. It is important to show the DIAPH3/DIAPH1 double KD. Following the logic of the paper this should suppress the furrow defects seen in DIAPH3 siRNA. However, it is possible that DIAPH1 substitutes for DIAPH3 in the siRNA condition. Please also show DIAPH1 and DIAPH3 localizations in the absence of the other.

Response: The reviewer raises an interesting and logical extension of our model predictions. However, simultaneous depletion of both DIAPH1 and DIAPH3 significantly reduces cell viability to the point where we have no viable mitotic cells to analyze. We have quantified these effects and they are shown in reviewer Fig 3.

With respect to the reviewer's model of DIAPH1 and 3 redundancy due to the re-localization of one in the absence of the other, this does not occur. Upon depletion of DIAPH1, DIAPH3 remains at the furrow and conversely upon depletion of DIAPH3, DIAPH1 remains evenly distributed around the plasma membrane. This data is shown in Supplementary Fig 1e.

6. Fig3. In 3a please show how measurements were made using boxes.
 In 3c – why is there so much anaphase elongation with DIAPH3 siRNA and Pm DIAPH1?
 This suggests that there is a second cue that polarizes the cortex.
 If possible, please quantify the dynamics of the elongation at anaphase under these conditions.

Response: We have now included a box to outline where the Lifeact-GFP intensity measurements were made.

With respect to cell elongation we measured cell elongation rates in all conditions and added this data to Fig 1f and Supplementary Fig 1. Our data suggest both actin isoforms networks have some role in cell elongation. At present we do not know if each has a direct or indirect role, or whether they act independently. Further work beyond the scope of this study will be required to make concrete assertions. The DIAPH3 siRNA – pm-GFP-DIAPH1 expressing cells do not elongate greatly, as shown in the graphs that quantify cell elongation. In the original panel within Fig 3c, the DIAPH3 siRNA-pmDIAPH1 cells were of slightly greater magnification than that of the other panels in Fig 3c to allow the reader a more detailed view of the cell to convince the reader that no furrow was observed. However, this may have given the false impression of greater cell elongation than actually occurred (see Fig 1f and Supplementary Fig 1). In the revised submission we have adjusted the magnification of the DIAPH3 siRNA-pmDIAPH1 panel to one common with the other panels in the figure.

7. The authors state "As astral microtubules are required for polar relaxation during anaphase". Please show cells with no centrosomes and present the extent of anaphase elongation to make the extent of the defect clear in their system. It is very unlikely to be as black and white as this.

Response: We have endeavoured to generate acentrosomal HeLa cells, however the experiment is technically difficult and requires conditions that severely compromise the cells. Through careful variation of conditions, we derived a protocol that required 500nM Centrinone B for at least 72 hours. This treatment strategy utilizes a 5-fold higher concentration of Centrinone B over a longer period of time. Any higher concentrations killed all cells. Even in this condition approximately 25% of the cells died suggesting the cells were under considerable stress. Using centrosome and spindle pole markers, we could find cells lacking centrosomes. However, much to our surprise, when cells entered anaphase there were numerous non-spindle microtubules remaining many close to the polar cortex of anaphase the cells (refer to reviewer figure 4 below). We are uncertain of the nature, origin and dynamics of these microtubules. These observations are in marked contrast to what we documented in the mono-centrosomal cells reported in the manuscript (Fig 5) where the cytoplasm between the spindle pole lacking a centrosome and the polar cortex is almost completely devoid of microtubules. We do not know how these microtubules are formed. While we do observe phenotypes consistent with our other data in the manuscript, including a reduction in γ -actin clearance from the poles (γ -actin intensity ratio pole / furrow of 1.22 compared to control 0.97) and a reduction in cell elongation rates that are comparable to expression of a constitutively active DIAPH1. However, the presence of microtubules near the polar cortex despite the absence of centrosomes complicates the interpretation. Therefore, we are not confident in being able to draw firm conclusions attributable to a lack of microtubules communicating with the polar cortex. In light of these experimental limitations we have amended the text to “as astral microtubules are implicated in polar relaxation” and cited references from previous studies.

8. In fig 3c, both DIAPH3 and DIAPH1 appear to have blebs at metaphase – please quantify.

Response: We have now quantified the metaphase blebbing in all conditions and added this data to Fig. 1e. The data shows that little blebbing occurs in metaphase, however there is a large increase during anaphase and this is amplified upon DIAPH1 depletion.

9. If the author's model is correct, why does the cortex clear on the side far from astral MTs in monopolar human cells forced to leave mitosis? This is an important issue to address and to discuss in the paper. It would be great if the authors could reverse this, e.g. in a DIAPH3 KD.

Response: We are uncertain which images the reviewer is referring to in making these conclusions, as we see no evidence for this. We have re-annotated figures in the hope of clarifying them. In all cases the actin does not clear from the pole of the cell lacking astral microtubules. Clearing only occurs at the pole with microtubules. These observations are consistent with our model of astral microtubules delivering an inhibitor of actin nucleation to the

poles of the cell, thereby facilitating the loss of actin from the pole. This model predicts that in the absence of astral microtubules actin should persist at the pole, which is indeed what we observe.

Response to "Specific comments":

1. *While the focus is on human cells, it would be good to comment on other systems (flies and worms). How would findings made studying β and γ actin / DIAPH1 and DIAPH3 isoforms play out in organisms where cytokinesis is likely to be mediated by a single actin and Dia homolog?*

Response: It is difficult to make any firm predictions as we are not aware of reagents that would enable the different actin isoforms of invertebrate organisms to be assessed. All metazoans we have examined have multiple actin isoform genes and all have multiple formin genes. Based on work in mammals it would be expected that these other formins and actins have roles, but in the absence of data we are reluctant to comment either way. Generating specific reagents and tools for invertebrates would be excellent future directions.

2. *There is no mention of Arp2/3. This has been shown to play an important role in the HeLa cortex and in anaphase in many systems. How much of the γ actin is nucleated downstream of Arp2/3?*

Response: Whilst we agree that Arp2/3 is a very important actin nucleator a previous study by Bovellan *et al* 2014 *Curr. Biol.* found little role for Arp2/3 during cytokinesis using the CK666 inhibitor in HeLa cells. We repeated those experiments and obtained the same data. Consequently, we focused on the formins.

3. *The discussion on blebs ignores the fact that they are sites of actin and Myosin II recruitment. How do the authors imagine this happens at cell poles? Note, a loss of membrane blebs can occur with a uniform gain of cortical tension as well as a loss. Blebs happen when there is an imbalance of forces.*

Response: We would expect blebs to be retracted in anaphase the same way they in other phase of the cell cycle, by the recruitment of actin, actin nucleators and myosin II after the bleb has formed. Perhaps if astral microtubules do not invade the bleb, then DIAPH1 can be recruited to the plasma membrane of the bleb, producing actin that recruits myosin II and facilitates bleb re-traction. However, this is significant conjecture and far beyond the scope of our data and would be more suitable to discussion after further study focused directly on that question.

4. *The authors state "As anaphase proceeded, we observed that the Lifeact-GFP signal decreased at the cell poles (Fig. 3a) due to the cortical flow of actin from the poles to the equator". The authors shouldn't assume this is true without measuring it. Temper the language here, i.e. remove 'due to'.*

Response: The reviewer is quite correct and we have tempered the language in the revised manuscript.

5. The authors state "Surprisingly, the loss of the β -actin contractile ring upon depletion of DIAPH3 or anillin, an enhancer of DIAPH3, also allowed a rudimentary, although unstable furrow to ingress (Fig. 3b, c, Supplementary Fig. 2, Supplementary video 4) 9,12,18." This is likely due to DIAPH1. Test and/or discuss.

Response: We have amended the wording and discussed this point. In none of our micrographs do we see an enhanced recruitment of DIAPH1 or γ -actin to the cell equator or nascent furrow upon DIAPH3 depletion (See supplementary Fig 1e), suggesting that the nascent furrow we and others observe is not due to DIAPH1 recruitment to the cell equator.

6. the authors state: "As astral microtubules are required for polar relaxation during anaphase..." Again, show this or temper the language.

Response: We have tempered the language.

7. It would be good to add a label 5a Rho (which Ab?)

Response: We have clarified the labelling of Rho in Fig 5a and the source of the antibody is in the methods and material section.

Response to reviewer 2

The manuscript authored by Chen et al. describes the role of β - and γ -actin in cytokinesis. By using time-lapse fluorescence observation of actins, formins and GAPs, they concluded that β -actin network is formed at the cleavage furrow mainly by DIAPH3, whereas γ -actin localized at the cortex except for cell poles, which is caused by deactivation of DIAPH1. They also utilized atomic force microscopy (AFM) to measure the stiffness of the cell cortex during mitosis. Mechanical characterization of mitotic cells by AFM has been conducted by a number of previous studies, and a well-established approach. However, the measurement and the analysis requires highly careful considerations depending on what is to be measured. Unfortunately, the experimental procedures of the AFM experiments in this study were not well-described in the manuscript, and therefore difficult to understand. The followings are the points that the authors should clarify.

i) In figure 2, the authors tried to compare "membrane tension" in normal cells and formin-knock-down cells to explain delocalized and increased membrane blebbing they observed in figure 1. I agree that the membrane blebbing largely depends on a local membrane tension, which is determined not only by osmotic pressure but also interaction between cortical skeleton and the lipid bilayer. Therefore, measuring membrane tension in various knock-down cells may provide a useful information. However, on the other hand, the experimental approach that they utilized to measure the "membrane tension" seems inappropriate. The authors claimed they measured "membrane tension" by recording and analyzing the force-indentation curve of AFM cantilever. However, the force curve obtained by pushing the cantilever against the cell surface contains the stiffness of both plasma membrane and cortical skeleton. Since the cortical skeleton

(cortical actin) is much stiffer than the plasma membrane, the Young's modulus obtained from the force curve mostly indicates the stiffness (elasticity) of the cortex and not that of the plasma membrane. Since the authors did not describe the detail procedure of the force measurement (loading rate, indentation depth, etc.) it is hard to understand how they extracted the information of membrane tension and exclude the stiffness of the cortex. Or, do they want to measure the elasticity of the cortical skeleton? If so, how they link it to the blebbing? The authors have to describe what they want to measure (membrane tension or elasticity of the cortex) by AFM, and show the detail procedure to extract the information they want from the obtained force curve.

Response: We agree with the Reviewer that we are in fact measuring both cortical skeletal tension as well as plasma membrane tension in our studies, as the plasma membrane and actin cortex are linked. In general, however, the contribution of membrane tension to overall cell surface tension is minimal in mitotic cells (see Taubenberger *et al* 2020 Front Cell Dev Biol. and Kelkar *et al* Curr. Opin. Cell Biol. 2020). Thus, our measurements primarily reflect the tension of the cortical cytoskeleton, which additionally correlate well with cortical cytoskeleton stiffness (see Taubenberger *et al* 2020 Front. Cell Dev. Biol.). Importantly, and as we show in the manuscript, focal changes in cytoskeletal tension can lead to localized structural alterations of the overlying plasma membrane (such as blebbing). We have modified the text accordingly

We have included additional detail in the Methods section regarding how we analyzed the force curves generated by our AFM measurements. We have now expanded the relevant text in the Methods section as requested.

ii) The authors described that they obtained the force-indentation curve from metaphase cells. I wonder why they used metaphase cells and not the telophase and cells in cytokinesis, which is the most important steps of actin and formin function. They showed different distributions of β - and γ -actins, as well as DIAPH1 and 3, during cytokinesis (figure 1), but not in metaphase. It has been known that the properties of the cell cortex drastically changes upon the transition from metaphase to anaphase. So, I wonder why they measured the elasticity of metaphase cells, and not in the following period, which is highly confusing. The authors should clarify this point. If possible, they should perform the same type of force measurement with telophase cells or cells in cytokinesis.

Response: Our goal was to obtain some measurements with respect to cortical stiffness upon disrupting the actin cytoskeletons. Often in the literature a statement is made about cortical stiffness or membrane tension without any measurement being made, especially with respect to observing blebs. Ideally, as the reviewer suggests, we sought to make measurements during anaphase. However, due to the magnitude of blebbing in some disrupted conditions (eg DIAPH1 or DIAHP3 depletion) it was not possible to make measurements. In order to make comparisons between experimental conditions, we therefore chose to perform all measurements in metaphase. We have amended the text to clarify this. We are unable, due to COVID lockdown restrictions, to attempt further AFM experiments as the equipment is located in another building to which we do not have sufficient access.

iii) As mentioned above, the description of the experimental procedure of AFM measurement is insufficient. They just cited a paper (Santoro R., et al.) in which the elasticity of interphase cells were measured by AFM. However, since the mechanical properties of the adherent cells in

interphase is different from that of mitotic cells, they have to optimize the procedures of the measurement and analysis. Especially, if they want to measure the “membrane tension”, the indentation depth and the way of curve fitting is critical (in the cited paper, indentation depth was 500 nm, but this is too far to measure the properties of the plasma membrane). I strongly recommend the authors should show raw data of force curve and explain how they fit it with Hertzian model. It may be necessary to show representative force curves from individual knock-down cells as supplementary.

Response: We have added a fuller and more detailed description of our AFM procedures and have also added forces curves.

We agree with the Reviewer that the mechanical properties of cells change as they enter mitosis. The methods we employed to measure cortical cytoskeleton/plasma membrane tension have been used in the past in mitotic cells (see Kunda *et al* Curr. Biol. 2008, Matzke *et al* Nat. Cell Biol. 2001 and Matthews *et al* Dev. Cell 2020). We have now included these references in an expanded methods section. The Hertzian model is the standard model used in the literature to determine the mechanical properties of biological or biomaterial samples, and is commonly used to calculate cortical cytoskeleton/plasma membrane tension between two spherical elastic bodies. This model can be used when the indenter shape is spherical and the sample is spherical or hemispherical, such as a cell.

As requested, we have now included a detailed description of the methods used to analyze the AFM-derived force-displacement curves in the Methods section. We have also included representative force curves as Supplementary Figure 3 from individual knockdown cells.

Response to reviewer 3

In this remarkable and excellent study, the authors provide multiple unique and fundamental novel insights into how γ -actin at the poles of anaphase cells is regulated to permit efficient cytokinesis. They first establish the differential distribution of β - and γ -actin, at the furrow and along the cortex respectively during anaphase, and the related formins DIAPH1 and DIAPH3. Disruption of DIAPH1 or its activator IQGAP1 induce the loss of cortical γ -actin, blebbing around the anaphase cells and increase in multi-nucleated cells indicative of cytokinesis failure. Cells lacking function DIAPH1 are much less stiff as assessed by AFM, as expected. Next, they show that appropriate furrow ingression requires actin assembly by both DIAPH1 and DIAPH3. In a nice experiment following treatment with centrinone B to limit Aurora-A to just one spindle pole body and so limit astral microtubules to that pole, they show that DIAPH1 and γ -actin removal at the pole requires astral microtubules. Next, they explore the mechanism of DIAPH1 inhibition at the pole. These formins are regulated by an N-/C-terminal inhibitory interaction. RhoA or IQGAP binding to the N-terminal region of DIAPH1 can abolish its ability to bind the catalytic C-terminal region. Based on studies that showed the activator IQGAP1 can bind the microtubule associated protein CLIP170, they find that CLIP170 can sequester IQGAP1 from DIAPH1 and thereby remove the actiavtor. Finally, using proximity ligation assays they show that in metaphase, IQGAP1 associated with DIAPH1 at the cell cortex and then with CLIP170 in anaphase, indicating that the delivery of CLIP170 by astral microtubules locally deactivates DIAPH1 at the poles, and this contributes to the formation of the cleavage furrow. Building on their 2017 Nat. Comm. paper, this provides many notable advances to very long-

standing issues: the clearest functional distinction between β - and γ -actin; two related formins that drive assembly of selected actin isoforms; regulation of both cortical γ -actin and cytokinetic β -actin are needed for furrow ingression; that CLIP170 can remove the formin activator IQGAP1; this regulation is mediated in a temporal and spatial manner by astral microtubules. This is one of the best papers I have read in a long time - superior in results and message to many in the major CNS journals.

My only comment relates to the distribution of γ -actin. Wouldn't one expect γ -actin to be diminished from the poles (as diagrammed in Fig. 8) at late anaphase in Fig. 1A and Fig. 4a control and as reported by others (eg. Rodrigues et al. Nature 524, 489)? In the centrinone B experiment, it seems it is diminished where there are astral microtubules. A statement about this would clarify the issue.

Response: We expect and see γ -actin depletion at the poles where there are astral microtubules. Whilst it may not be as striking in the γ -actin staining image as the Lifeact-GFP we have added further quantitation that demonstrates this loss of γ -actin at the poles.

REVIEWER COMMENTS

Reviewer #1 (Remarks to the Author):

We congratulate the authors on their revision. The revised manuscript is much more complete. It is an excellent piece of work provides an important new set of data supporting a role for MTs in the disassembly of the polar actin cortex in cytokinesis.

The authors have also clarified many points of confusion and have explained why some experiments are technically challenging and therefore not possible and where necessary have provided alternatives.

The answers to the reviewers' concerns are satisfactory, with only a few exceptions which we would ask the authors to look at before publishing:

1. The authors data point to a role for Rho and DIAPH1 in the assembly of a mechanically rigid metaphase cortex that is remodelled in anaphase to drive polar blebbing, cell elongation and division. This is not really made clear in the text (e.g. in the model in Figure 8), which suggests that Rho and Formins are only activated at anaphase.

e.g. "102 DIAPH1 localizes to the cell cortex during anaphase along with γ -actin filaments (Fig. 1a), "

This doesn't fit with the cell biological (e.g. PLA data in Fig 7b) or mechanical data presented (which only show metaphase) in the paper. Thus, it would be good if the text and model were modified to make clear what changes from metaphase to anaphase.

We note here that the authors do not show mechanical data for DIAPH3 siRNA cells as was requested in the last round of reviews. Was this done? If not, is DIAPH3 cortically localised prior to anaphase? If so, is the expectation that DIAPH3 contributes to metaphase cell stiffness? The data and/or any uncertainties in this regard should be made clear in the manuscript.

2. The authors suggest that DIAPH1 and DIAPH3 are functionally independent. Clearly this is not the case (Fig 1c). The authors should make it clear in the text precisely how the localisation of each formin depends on the other.

3. The authors still suggest without clear evidence that CLIP170 delivery is polarised at anaphase.

"As expected, GFP-CLIP170 was concentrated on spindle microtubules in metaphase, but from 252 the onset of anaphase onwards, GFP-CLIP170 began to transit towards the polar cortex (Fig. 7a, 253 Supplementary video 6). "

This is not clear from the data. The statement should be removed and/or toned down or new data provided.

4. Finally, the authors did not understand the question about monopolar exit. We apologise if we weren't clear on this point. In this we were referring to the many papers that have explored cortical remodelling in cells arrested in mitosis that have been forced to exit mitosis following the addition of CDK inhibitors (note that this type of approach would enable the authors to study any conditions in

which cells enter mitosis but do not exit, e.g. in CLIP170 siRNA cells). In these papers, e.g. Hu et al., 2008, cells polarise as they exit mitosis so that the DNA and both centrosomes are found at one pole (where cortical actin is lost) while a distal actomyosin cortex is assembled at the other pole. In some cases, this actomyosin cortical assembly has been associated with long MTs, in other cases it does not appear to require microtubules at all. It would be good if the authors could discuss this in the manuscript and explain how they think their data fits with the data from many groups that have studied this type of forced exit.

Reviewer #2 (Remarks to the Author):

In the revised manuscript, I found a great improvement in the description of the force measurement by AFM.

On the other hand, the validity of measuring metaphase cells instead of anaphase cells (comment No. ii) has not been improved. The authors agreed that the properties of the cortex should be characterized in anaphase cells. However, they did not perform the experiments due to technical difficulty and inaccessibility of the AFM facility.

As I explained in the first review, the mechanical properties of the cell cortex in metaphase and anaphase cells are significantly different. Therefore, I strongly believe that the results from anaphase cells will greatly strengthen their conclusion. The mechanical properties of the cell cortex throughout mitosis can be measurement by AFM using tip-less cantilever [Stewart MP, Nat. Method, 2012]. Therefore, I would like to suggest that the authors perform the force measurement on anaphase cells when their AFM facility becomes accessible.

Reviewer #3 (Remarks to the Author):

As I said at the outset, this is a very nice study that should be accepted.

Response to Reviewers

Changes to the manuscript text have been highlighted in yellow.

Reviewer #1 (Remarks to the Author):

We congratulate the authors on their revision. The revised manuscript is much more complete. It is an excellent piece of work provides an important new set of data supporting a role for MTs in the disassembly of the polar actin cortex in cytokinesis.

The authors have also clarified many points of confusion and have explained why some experiments are technically challenging and therefore not possible and where necessary have provided alternatives.

The answers to the reviewers' concerns are satisfactory, with only a few exceptions which we would ask the authors to look at before publishing:

1. *The authors data point to a role for Rho and DIAPH1 in the assembly of a mechanically rigid metaphase cortex that is remodelled in anaphase to drive polar blebbing, cell elongation and division. This is not really made clear in the text (e.g. in the model in Figure 8), which suggests that Rho and Formins are only activated at anaphase.*

e.g. "102 DIAPH1 localizes to the cell cortex during anaphase along with γ -actin filaments (Fig. 1a), "

This doesn't fit with the cell biological (e.g. PLA data in Fig 7b) or mechanical data presented (which only show metaphase) in the paper. Thus, it would be good if the text and model were modified to make clear what changes from metaphase to anaphase.

Response: We have clarified in the text and in particular in Fig 8 that we expect, based on numerous published studies, that Rho and actin nucleators are active during metaphase. Fig 8 now depicts the relationship of Rho, DIAPH1, DIAPH3 β -actin and γ -actin in metaphase to further clarify this and reflect our localization data in Fig 1a, Fig 5a and Sup. Fig 5. This shows that all of these factors are present at the cortex and are expected to be active (as the different actin isoform networks are present) during metaphase. In contrast, as the cells progress into anaphase, we describe a redistribution of these factors . We have shown this directly throughout the manuscript as well as in our previous studies (Chen et al 2017).

Our statement that "DIAPH1 localizes to the cell cortex during anaphase along with γ -actin filaments" is correct as it accurately describes our data in Fig 1a to which it directly refers. The IF data in Fig 1a (and supported by IF data in Sup Fig 1c) clearly shows DIAPH1 and γ -actin on the cortex during anaphase. We have now prefaced the statement with a sentence describing the distribution of DIAPH1, 3, β and γ -actin in metaphase, which is also depicted in Fig 1a.

It should be noted that PLA does not reflect simple co-localization, rather it reflects the proximity of two ligands and therefore infers interaction. The absence of PLA signal does not mean that each ligand is absent from a site, merely it demonstrates the two ligands are not in close proximity to each other i.e. are not interacting. Therefore, there is nothing in Fig 7b that contradicts the statement “*DIAPH1 localizes to the cell cortex during anaphase along with γ -actin filaments*”.

We note here that the authors do not show mechanical data for DIAPH3 siRNA cells as was requested in the last round of reviews. Was this done? If not, is DIAPH3 cortically localised prior to anaphase? If so, is the expectation that DIAPH3 contributes to metaphase cell stiffness? The data and/or any uncertainties in this regard should be made clear in the manuscript.

Response: AFM of DIAPH3 depleted cells cannot be performed as we do not have access to the AFM equipment due to COVID19 pandemic restrictions on laboratory usage as discussed in our previous response to reviewers.

With respect to DIAPH3 localization prior to metaphase, Fig 1a in the current and previous manuscript submissions clearly demonstrate that DIAPH3 localized over the whole cortex during metaphase. As stated in the manuscript, we cannot rule out that DIAPH3 in some way contributes to mitotic cortical stiffness, indeed we may expect it, but are reluctant to comment without data. However, we note that the focus of our study is anaphase and cytokinesis and not metaphase. We agree that an analysis of factors involved in generating cortical stiffness during cell rounding and metaphase would be an interesting future study, but it is beyond the scope of the work we present here. We feel that the question merits its own study where all factors and pathways can be interrogated in detail.

2. *The authors suggest that DIAPH1 and DIAPH3 are functionally independent. Clearly this is not the case (Fig 1c). The authors should make it clear in the text precisely how the localisation of each formin depends on the other.*

Response: We respectfully disagree with the reviewer as we have no evidence functional dependence between DIAPH1 and 3, all our evidence points to them being functionally independent during cytokinesis.

Fig 1c concerns the relationship of DIAPH1 to β and γ -actin localization. It demonstrates that depletion of DIAPH1 or removal of DIAPH1 actin nucleating activity (the I862A mutant) disrupts the accumulation of γ -actin at the cortex, but does not affect β -actin accumulation at the furrow, therefore demonstrating DIAPH1 does not regulate β -actin during cytokinesis. In contrast, we have previously demonstrated (Chen et al 2017) that DIAPH3 depletion disrupts β -actin accumulation at the furrow but not γ -actin at the cortex, thereby demonstrating that DIAPH3 does not regulate γ -actin during cytokinesis.

Our assertion that DIAPH1 and DIAPH3 are functionally independent is derived from the data described above and the following additional data.

1. In cells where DIAPH1 was depleted by siRNA, DIAPH3 and β -actin still localizes to the nascent furrow (Fig 1c and Sup Fig 1e)

2. In cells where DIAPH3 was depleted by siRNA, DIAPH1 and γ -actin still localizes to the nascent furrow (Sup Fig 1e, Chen et al 2017)
3. DIAPH1 is activated by an interaction with IQGAP1 and Rho-GTP (Chen et al 2020).
4. DIAPH3 is activated by an interaction with anillin and Rho-GTP (Chen et al 2017).
5. DIAPH1 does not interact with anillin (Chen et al 2020 and our current study Sup Fig 5f).
6. DIAPH3 does not interact with IQGAP1 (Chen et al 2020 and our current study Sup Fig 5f). These data (points 5 and 6) demonstrate that DIAPH1 and DIAPH3 are activated by two different pathways.
7. DIAPH3 is recruited to the furrow by anillin (Watanabe, et al 2010 and Chen et al 2017).
8. DIAPH1 is recruited to the cortex by IQGAP1 (Brandt et al 2007, Chen et al 2020). The data in points 6 and 7 demonstrate that DIAPH1 and 3 are targeted to the plasma membrane through different molecular interactions.
9. DIAPH1 and IQGAP1 interact at the cortex in metaphase and this decreases during anaphase (this study, Fig 7b).

Combined the data from our studies and those of others strongly support the conclusion that DIAPH1 and DIAPH3 act independently. We have no evidence of inter-dependence between DIAPH1 and 3.

3. *The authors still suggest without clear evidence that CLIP170 delivery is polarised at anaphase.*

“As expected, GFP-CLIP170 was concentrated on spindle microtubules in metaphase, but from 252 the onset of anaphase onwards, GFP-CLIP170 began to transit towards the polar cortex (Fig. 7a, 253 Supplementary video 6). “

This is not clear from the data. The statement should be removed and/or toned down or new data provided.

Response: We do not state, nor have we in previous submissions, that CLIP170 delivery is polarized, this is an inference of the reviewer that is not borne out by the data. We assert, based on our observations and the data provided, that little or no CLIP170 is observed on astral microtubules in metaphase. In contrast, CLIP170 is abundantly observed on astral microtubules in anaphase. We have further clarified this in the text.

The data, movies and still images including PLA data, clearly demonstrate that during metaphase we are unable to detect CLIP170 on astral microtubules growing toward the cortex nor directly interacting with cortically localized IQGAP1. During metaphase we only detect CLIP170 on spindle microtubules. In contrast, during anaphase we now detect CLIP170 on astral microtubules growing toward the polar cortex and CLIP170 on spindle midzone microtubules. The statements in the manuscript reflect our observations and are now further qualified by stating that in anaphase we see CLIP170 on astral microtubules growing toward the polar cortex and in spindle midzone microtubules between the segregating chromosomes.

4. *Finally, the authors did not understand the question about monopolar exit. We apologise if we weren't clear on this point. In this we were referring to the many papers that have explored cortical remodelling in cells arrested in mitosis that have been forced to exit mitosis following the addition of CDK inhibitors (note that this type of approach would enable the authors to study any conditions in which cells enter mitosis but do not exit, e.g. in CLIP170 siRNA cells). In these papers, e.g. Hu et al., 2008, cells polarise as they exit mitosis so that the DNA and both centrosomes are found at one pole (where cortical actin is lost) while a distal actomyosin cortex is assembled at the other pole. In some cases, this actomyosin cortical assembly has been associated with long MTs, in other cases it does not appear to require microtubules at all. It would be good if the authors could discuss this in the manuscript and explain how they think their data fits with the data from many groups that have studied this type of forced exit.*

Response: We have expanded our previous discussion in the manuscript with respect to suggesting two distinct populations of microtubules with different molecular characteristics existing within the cell during metaphase exit. Such a concept could explain why two different populations of microtubules exert two differing spatially restricted effects on the actin cytoskeleton. However, we are reluctant to speculate further in the absence of investigating the molecular characteristics of these microtubule population, which is beyond the scope of this study. We feel that a much more extensive experimental characterization will be required for us to make an insightful vs. wildly speculative contribution.

With regards to the reviewer's specific comments, the data in Hu et al point to two populations of MTs—those that emanate from the centrosomes to the cell cortex where there is a coincident depletion of actin and others that appear to emanate from near the chromosomes and extend to the opposite pole of the cell to promote actin polymerization. This would appear to be similar to the situation reported by Canman et al in 2003 where they proposed a model “in which chromosomes supply microtubules with factors that promote microtubules stability and furrowing”. We speculate in the text that what the authors see are the differential effects of astral vs spindle midzone MTs, where the long microtubules extending through the region of the chromosomal mass to instigate furrowing are analogous to spindle midzone microtubules that perform the same function in wildtype cells. What the molecular difference between these two microtubule populations is we do not know and is beyond our discussion, which focuses predominantly on astral microtubules that are the focus of our experimental data.

Reviewer 2

In the revised manuscript, I found a great improvement in the description of the force measurement by AFM.

On the other hand, the validity of measuring metaphase cells instead of anaphase cells (comment No. ii) has not been improved. The authors agreed that the properties of the cortex should be characterized in anaphase cells. However, they did not perform the experiments due to technical difficulty and inaccessibility of the AFM facility.

As I explained in the first review, the mechanical properties of the cell cortex in metaphase and anaphase cells are significantly different. Therefore, I strongly believe that the results from

anaphase cells will greatly strengthen their conclusion. The mechanical properties of the cell cortex throughout mitosis can be measurement by AFM using tip-less cantilever [Stewart MP, Nat. Method, 2012]. Therefore, I would like to suggest that the authors perform the force measurement on anaphase cells when their AFM facility becomes accessible.

Response: Whilst we agree that additional strategies may allow AFM measurements in violently blebbing anaphase cells our lack of access to the atomic force microscope in a 3rd party lab will remain severely limited for many months during the on-going COVID pandemic. Waiting for access would put off publication for 9 months or more. We believe our finding that DIAPH1 contributes to cortical stiffness in metaphase and that its inhibition at cell poles in anaphase which is concomitant with loss of polar γ -actin and increased blebbing reflects a drop in polar cell stiffness still stands.

Reviewer 3

As I said at the outset, this is a very nice study that should be accepted.

Response: We thank reviewer 3 for their continued support of our work and the findings that stem from them.